

**More trees, more rain? The unexpected role of forest and aquifers on the global water cycle**
Jean Chéry[1], Michel Peyret[1], Cédric Champollion[1] and Bijan Mohammadi[2]
[1] Geosciences Montpellier, Montpellier University, France
[2] Institut de Mathématiques Alexandre Grothendieck, Montpellier University, France
Corresponding author: Jean Chéry, e-mail: jean.chery@umontpellier.fr
**Abstract**
Forests' influence on rainfall has been debated since antiquity, with historical observations suggesting
that deforestation reduces precipitation. While early scientists believed that forests attract rain, later
research provided conflicting views, and modern climate models remain inconclusive on forests' role in
regional precipitation at continental scale. A difficulty of global climate model is the need to integrate
processes associated to multiple scientific field, like meteorology, hydrology and forest ecology. As a
result, the numerous constitutive equations, the complexity of numerical codes and the variability of
natural situations make difficult to provide a quantitative water cycle model controlled by a small
number of key parameters. Assuming water mass conservation between ocean, atmosphere, ecosphere
and continental aquifers, we revisit this problem and we build a simplified physical scheme of water
transfer across these reservoirs. Assembling a limited number of constitutive equations into a numerical
code, we propose a parsimony model able to simulate rainfall distribution patterns. Starting from initially
different vegetation conditions (sparse and dense), we identify the parameters modulating steady state
continental precipitation for both situations. First, we show that sparse vegetation patterns do not lead
to uniform and large continental precipitation. Rather, continental moisture is spatially limited by
atmospheric dispersion and wind, thus providing a decay of precipitation as a function of coastal distance
as observed in most cases worldwide. Second, our model predicts that large and widespread continental
precipitation results from the combination of dense vegetation patterns and deep aquifers low hydraulic
conductivity. This ecological/hydrological interaction is due to the slowness of horizontal water flow,
allowing underground reservoir level to increase enough for reaching plant roots, inducing in turn
vegetation evapotranspiration and atmospheric moisture. Overall, our model shed new light on the
balance between atmospheric water vapor transport by wind and dispersion, vegetation
evapotranspiration and underground water flow. Noticeably, the efficiency of the water cycle and its
related precipitation distribution not only depends from atmospheric parameters but is also largely
modulated by vegetation dynamics and underground aquifers behavior. We finally discuss how forest
growth or destruction may alter continental precipitation at various time scales.
"*Simplicity is the ultimate sophistication*" Leonardo da Vinci
"*In philosophy, Occam's razor (novacula Occami) is the problem-solving principle that recommends
searching for explanations constructed with the smallest possible set of elements. It is also known as
the principle of parsimony or the law of parsimony.*"
From Wikipedia
**1. Introduction**
1.1 Historical background
The possible influence of forests on the amount of precipitation has been debated since the Greek
Antiquity (Andréassian, 2004). Later, Christopher Columbus' son Fernando has related that his father
knew "*from experience*" that the disappearance of the forest that originally covered the Canary Islands,
Madeira and the Azores had reduced fog and rainfall. Likewise, he believed that the afternoon rains that
occur in Jamaica and the West Indies were the consequence of the islands "*lush forests*".



In 1787, Bernardin de Saint Pierre (cited in Descroix et al., 2001) notices about Mauritius island
in Indian Ocean that « *the vegetal attraction of the forests is so well matched by the metallic attraction*
*of the peaks of its mountains, that a field situated in an open area, in the neighbourhood, often lacks*
*rain, whereas it rains almost all year round in the woods which are not within rifle range*. » The idea
that forests attract and induce rain grew in France during the following decades and Dausse stated in
1842 that "*man has the power to change a wet climate into a dry one in just a few years by clearing*
*forests* ". Later in the 19th century, divergent views emerged and the opposite conclusion was reached
by Cézanne in 1872: "*forests cannot significantly alter the amount of rainwater that falls in a river*
*basin*" (also cited in Descroix et al., 2001).
Since the dawn of worldwide remote sensing, global scale data gathering and massive computing
meteorology, the vast amount of accumulated knowledge did not lead the scientific community towards
a clear understanding of the possible retroaction of forests on the regional annual precipitation at basin
scale. On the one hand, this diversity of opinion is not surprising because this scientific question
encompasses very different fields: theoretical physics, forestry, climatology, or/and hydrogeology.
Apart from fundamental chemistry and physics concepts, these communities share a limited common
knowledge, thus rendering difficult the building of an integrated vegetation / atmosphere / hydrosphere
/ geosphere theory. On the other hand, this lack of agreement is somewhat intriguing nowadays because
forest / rain interaction is studied using advanced numerical climate packages in which most of relevant
constitutive equations are embedded, including all segments of continental water cycle: oceanic
evaporation, water vapor advection and transport to the continents, plant evapotranspiration and root
uptake, and ultimately water infiltration, deep storage and the way back to ocean by springs and rivers
(Gimeno at al. 2012).
Recently, the idea that forests induce rain gained new interest following the work of Makarieva
and Gorshkov (2007) (hereafter referred as MG2007). They propose that profiles, roughly perpendicular
to coastlines, across continents can be classified into two groups: (1) one shows a nearly constant
precipitation as a function of coastal distance in densely forested tropical and boreal areas, while (2)
another group displays a pronounced decay of rainfall from the coast towards inner continental zones
associated with bare soil or grasslands. MG2007 also propose a physical mechanism called "biotic
pump" explaining why forests are increasing pluviometry at regional scale. Briefly, the water vapor
induced by forest transpiration is prone to condensation during its adiabatic ascent in the lower
troposphere if convective conditions are met, leading to an atmospheric pressure drop over forested
areas. As a result, a pressure difference occurs between the inner continent and the overseas areas,
inducing a large-scale sea breeze importing moisture over forest that adds to the local moisture already
present, therefore increasing the precipitation probability. Major concerns also arose for the
sustainability of the Amazonian under hotter/drier climates and deforestation, with identified cascading
effects potentially leading to tipping points when climate/vegetation interactions will feed themselves
into an aridification spiral (Bochow and Boers, 2023; Sampaio et al., 2007; Wunderling et al., 2022)
Although the biotic pump explanation of MG2007 gained some popularity into scientific ecology
communities (Nobre, 2007; Sheil & Murdiyarso, 2009), the physical ground of this theory has been
disregarded by most climate modelling researchers. For example, Meesters et al. (2009) argued that the
physical approach of evaporation as provided by MG2007 was flawed and that high precipitation
occurring at sites located far from the ocean can be explained adequately by traditional theory, that
includes large scale advective processes (Coriolis forces, Hadley cells and monsoons), associated with
local re-evaporation process by the forest (Gimeno et al., 2012). Therefore, the dominant belief in
climate modeling research groups is that the effect of large forest over regional pluviometry is still
obscure (Douville et al., 2024).
While the capability of the forests to induce rain and change the meteorological regime over wide
continental areas remains an unsolved question, societal issues are raising under the pressure of
anthropogenic activities. First, the rise of average temperature since the beginning of the 20th century
increases evaporation of water bodies on land, the hydrologic pressure of underground waters and the
amount of evapotranspiration by forests. Therefore, if forests do not influence the precipitation regime,
their net effect negatively acts on water resources, suggesting that their expansion should be controlled



(Rambal, 2015; Asselin et al., 2024). Second, the amount of natural forests at global scale has been
decreasing for centuries (Hansen et al., 2013), due to the expansion of cultivated areas in developing
countries, widespread wood use (building and energy purposes among others), forest fires associated
with droughts or land use changes (Mouillot & Field, 2005). Third, the human water consumption is
increasing faster than the population growth (Serrano & Valbuena, 2017). The question of whether or
not large forests are our enemies (with pumping into hydrosystems and losing water vapor into the
atmosphere) or rather allies of mankind (in sustaining continental pluviometry) should be at the top of
many countries' agenda and a priority research program for international agencies. If the latter
hypothesis – postulating a positive forest influence on pluviometry – is correct, mankind will face a
negative feedback: continental water stock will unavoidably escape to ocean due to the syzygy of human
water consumption, global temperature rising and forested zones clearance (Steffen et al., 2015).
1.2 Need for a parsimony model
Why this question is not yet solved? A common answer could be that global water cycle,
continental pluviometry and forest ecology are controlled by many processes encompassing numerous
scientific fields, whose complexity and non-linearity render the associated mathematical problem non-
tractable, even for the most powerful codes (Vargas Godoy et al., 2021). To circumvent this difficulty,
we develop a parsimony model of the global water cycle involving only the very necessary concepts and
constitutive equations, following inspiration from current dynamic vegetation model simplifications
using eco-evolutionary optimality concepts (Harrison et al. 2021), or theoretical models catching
complex processes into few simplification rules (Dakos et al. 2024).
Some examples of parsimony models (also called low-complexity models) demonstrate their
power to enlighten our vision of complex systems in all scientific fields. One may cite Lorenz's
dynamical model showing the divergence of meteorology governing equations (Lorenz, 1963); Mc
Kenzie's model wrapping up in a couple of equations sedimentary basins motion dynamics (Mc Kenzie,
1978). The theory of plate tectonics (e.g. Morgan, 1968; Le Pichon, 1968) is probably the best example
of assembling unrelated geophysical fields into a coherent, planet scale framework. This integrative
process brought into a comprehensive view isolated research fields: mountains building processes,
thermal evolution of the lithosphere, divergence of oceanic plate motions and seismology. These
examples illustrate how a scientific maze can be sometimes transformed into a solvable puzzle.
In the context of our study, a built-in, low-complexity model will aim to provide long time scales
simulation by imposing mass conservation along the global water cycle across all components:
continental surface and groundwater, vegetation and atmosphere. Although our model will be not able
to predict small scale (temporal and spatial) dynamics, it will provide time dependent and asymptotic
dynamic of the water cycle. Thanks to its numerical efficiency, this parsimony approach will allow to
investigate the sensitivity of model parameter and various model setup changes. To setup a low-
complexity model of forest/rainfall interaction, we organize our study as follows: first, we re-examine
the data showing the correlation between the large natural forests and annual precipitation on the
geographical areas studied by MG2007 with a specific focus on Australia; secondly, we build a
simplified numerical model of the water cycle from ocean to continent; thirdly, we present some
numerical experiments in order to decipher the parameters controlling regional scale precipitation;
finally, we propose an integrated view of forest/water relationships.
**2.1 Precipitation and forest coverage**
We present here 10 profiles that have been studied by MG2007 and subsequent papers (Fig. 1a).
Two are related to largely forested areas in tropical areas (the Amazon and Congo basins). Five profiles
correspond to a partially forested zone: two in temperate zones (from the Appalachians to the inner US
Great Plains, and East of the Mississippi basin), and three in boreal areas (the Ob and Yenisey basins,
Siberia, and the McKenzie basin, Canada). Finally, the remaining three profiles cross mostly deforested
areas (West Africa along longitude 5°E, North-East China and Northern Australia).
In order to examine the spatial evolution of the mean precipitation along these transects, from
200km seaward towards the inner lands, we used satellite imagery and remote sensing associated to the



GIRAFE database (Niedorf et al., 2024). Although coarser than other databases for precipitation,
GIRAFE has the advantage of providing data both over continents and oceanic areas at 1° resolution.
Here, we computed the mean annual precipitation over the 2010 decade. Since the transects cover a
large range of precipitation conditions, for each transect we normalized the mean annual precipitations
with their value at the coast (Fig. 1b). Finally, the local forest coverage is estimated from the percent of
tree cover supplied by the Global Forest Change database at year 2000 (Hansen et al., 2013).

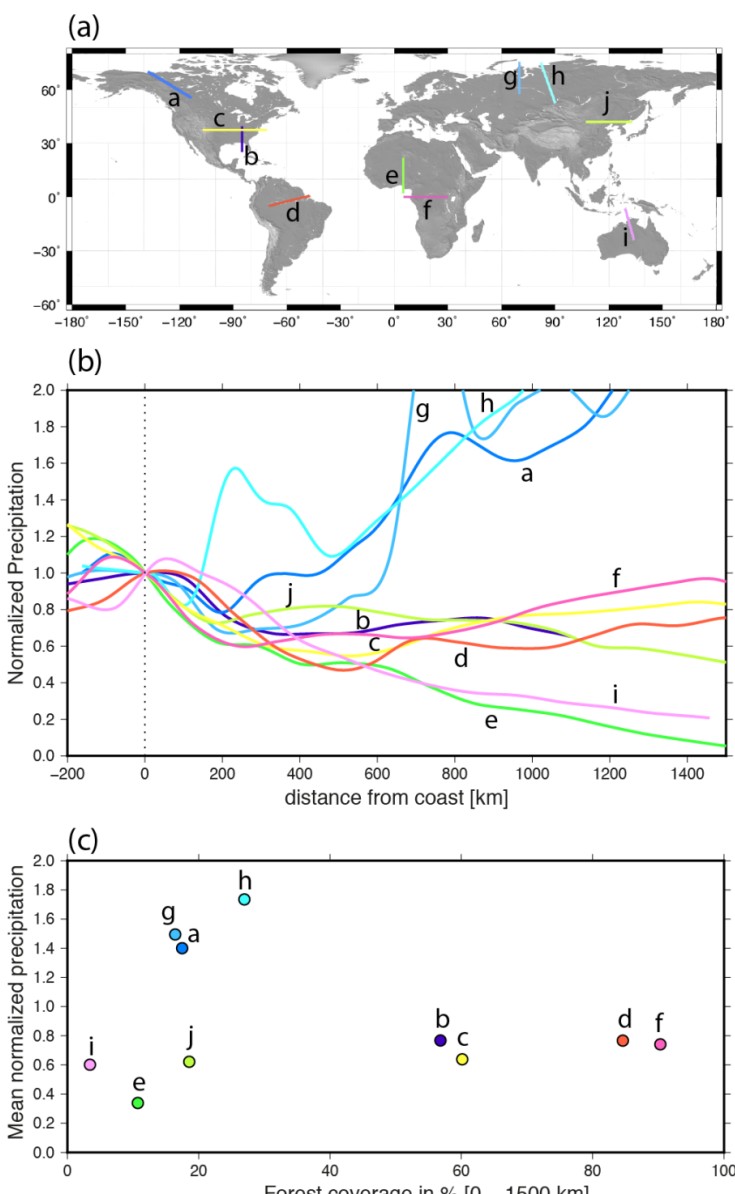

*Figure 1. MG2007 profiles revisited using mean annual precipitations estimated from the GIRAFE*
*database from 2010 to 2020. a) Geographical location of the 10 profiles. From west to east position of*
*the oceanic starting point of the transects, the labels correspond to the McKenzie basin (a), East of the*
*Mississippi basin at longitude 85°W (b), Eastern North America at latitude 37.5°N (c), the Amazon*





basin (d), West Africa at longitude 5°E (e), the Congo basin at latitude 0°N (f), the Ob basin (g), the Yenisey basin (h), North Australia (i) and East China at latitude 42°N (j). b) For each transect, the mean annual precipitation profile normalized by its value at the coast, as a function of distance to the coast. Negative distances correspond to the oceanic domain. Colors and labels on the right side of the figure identify the profiles. c) For each profile, we consider the mean annual precipitation over the 1500 first km on the continent side, normalized by the mean annual precipitation over the 200 first km on the oceanic side. These values are plotted as a function of the percentage of tree coverage over the first 1500 km determined from the Global Forest Change database.

A slight positive trend is observed between forest coverage and precipitation normalized to the ocean and most of these transects reveal that precipitation attenuation from ocean towards inner land is mitigated by the presence of forests. In extreme cases like Amazon or Congo basins, the annual precipitations as far as 1500 km inland are as high as along the coast (Fig. 1b-c). However, the correlation between forest coverage and rainfall is not as clear as claimed by MG2007. This is not surprising as many other factors contribute to pluviometry spatial distribution: atmospheric currents, troposphere temperature, daily and seasonal effects, orographic influences. For example, Ob, Yenisey and McKenzie profiles are clearly influenced by temperature increase from polar zone to cold temperate climate to the south, leading to significant southward increase of precipitation. Considering the numerous potential factors that may interact – even acknowledging a slight positive correlation between forested areas and precipitation (Fig. 1c) – it seems not possible to enlighten a causal relationship between forest and precipitation only on the base of observations.

To complement the previous data, we also examine the case of Australia, a continent isolated between Indian and Pacific oceans. Besides the GIRAFE and the Global Forest Change databases mentioned above, we extracted two additional physical values (evapotranspiration and relative humidity) from the JRA-55 climate dataset (Ebita et al., 2011). In the following, these values are averaged over two decades (2002-2021).

Australian vegetation is mostly coastal (east and north coast), the remaining part being covered by only sparse vegetation (Fig. 2a). Evapotranspiration distribution (Fig. 2b) is spatially associated with vegetation coverage and is also well correlated with average precipitation pattern (Fig. 3c). This map shows also that oceanic precipitations are intense to the north (monsoon effect) but also that the Pacific Ocean to the east is quite rainy (1000 mm/yr) with respect to Indian Ocean to the west at the same latitude (250 mm/yr). This latter effect might be the combination of several factors: (1) high oceanic humidity to the east, (2) high vegetation evapotranspiration along the eastern coast, (3) humidity advection by trade winds, and (4) atmospheric drying over the Australian central desert. Relative humidity (RH) pattern at ground level displays the same east/west asymmetry (Fig. 2d), also revealing an interesting feature: a systematical RH decreases from ocean to continent all around Australia, with damping distance ranging from almost zero on the west coast to ~600 km on the south east coast.




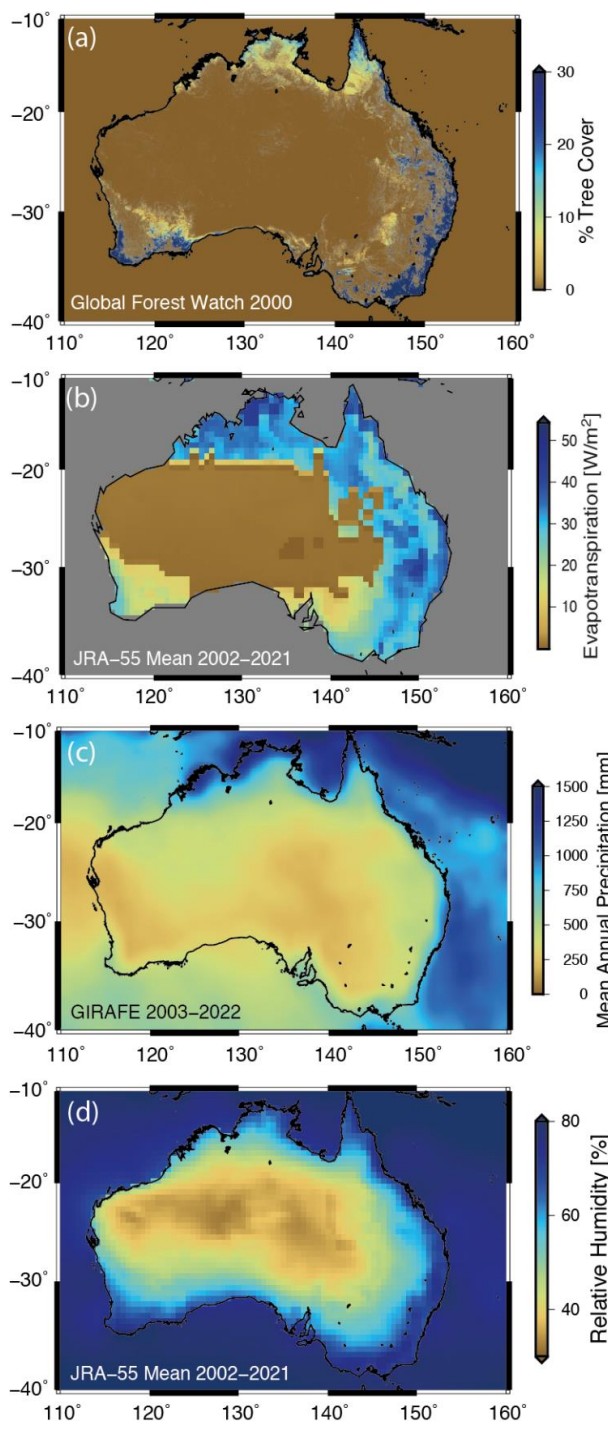

*Figure 2. Averages atmospheric and vegetation distribution on Australia (a) Tree cover percent in*
*2000 (Global Forest Change) (b) Averaged evapotranspiration 2002-2021 (JRA-55) (c) Averaged*
*precipitation 2003-2022(GIRAFE) (d) Average Relative humidity at 1000hPa 2002-2021(JRA-55)*



These atmospheric patterns suggest that oceanic moisture, its transport over the continent and
vegetation evapotranspiration combine in a complex way to provide precipitation pattern. This remark
is aligned with numerous research papers studying these processes interactions (e.g. Eltahir, 1998). At
this point, it should be reminded that the description above embraces only aerial water flow processes,
while underground water storage is of prime importance for feeding vegetation with water. Face to the
need to close water budget cycling back and forth between ocean and continent, we present an integrated
model approach of atmospheric and hydrologic water transport.
**3. Parsimony water cycle model**
Computing aerial and underground water transport involves three conservation equations for the
mass, momentum (or forces) and energy. As we are chiefly interested by water mass budget and long-
term climatic processes, we use mass conservation as a pivotal way to evaluate water motion during the
water cycle (Fig. 3). Momentum and energy equation are not solved, implying that variables like wind
velocity and temperature are directly included as input functions into water transport equations.

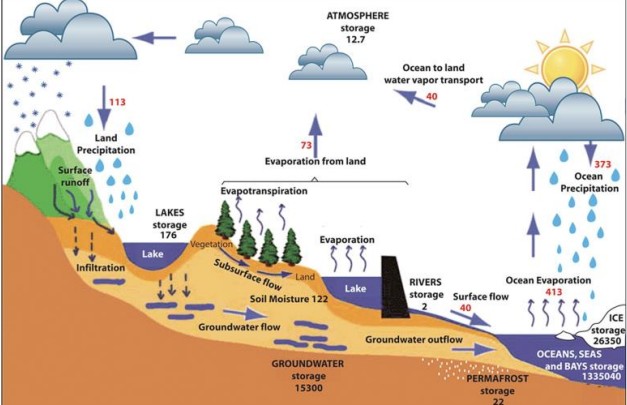

*Figure 3. hydrological cycle (from Gimeno et al., 2012).* Black numbers give estimates of the
observed main water reservoirs in $10^3$ km$^3$ and red numbers indicate the flow of moisture through the
system in $10^3$ km$^3$ yr$^{-1}$.
3.1 Tropospheric water vapor transport:
All flowing continental waters ultimately come from ocean evaporation carried by the
atmosphere. We assume that horizontal water transport mostly occurs as vapor into the lower
troposphere represented by a single layer having a thickness of about 5 km. We consider a 2D vertical
cross-section (Fig. 4), for which all physical variables represent average troposphere behavior, and the
tropospheric water vapor (WV) at a given point is governed by the following mass conservation
equation:
$\frac{\partial C}{\partial t} = div[-D_h grad_x(C) + vC] + \frac{\rho_w g}{P_{atm}}(\dot{E} - \dot{P})$                    (eq. 1)
where $C$ is water vapor concentration, $D_h$ is horizontal dispersion, $v$ is wind velocity, $\rho_w$ is water
density, $g$ is gravity acceleration, $P_{atm}$ is surface atmospheric pressure, $\dot{E}$ is evaporation/
evapotranspiration rate and $\dot{P}$ is precipitation rate (Table 1 provides units).Therefore, WV horizontal
transport is controlled by both concentration gradient and wind advection (see Rasmusson, 1968) for a




similar formulation). This equation converts a vertically integrated balance of WV into its concentration
(Gimeno et al. 2012, eq. 7). We also include a dispersion term that is generally not implemented in
simplified circulation models. Indeed, we conjecture that horizontal dispersion is of particular interest
for large scale WV distribution, as concentration gradients continuously sustain WV flow from wet to
dry areas. Apart from WV dispersion, effective horizontal dispersion can be estimated by combining
observation and modelling of long-range atmospheric transport of various products: nuclear products
dispersion (Ishikawa, 1995), volcanoes gas emission (Tiesi et al. 2006), pollutant plume transport (Pisso
et al. 2009), providing dispersion values ranging between $10^3$ and $10^5$ m²/s. We also consider water
sources and losses: evaporation (ocean) and evapotranspiration (plants) provide source terms $\dot{E}$ of eq.
1, while rain rate is associated to water loss $(-\dot{P})$. The sum of all terms of the right-hand side of eq. 1
corresponds to WV concentration rate change at a given location.

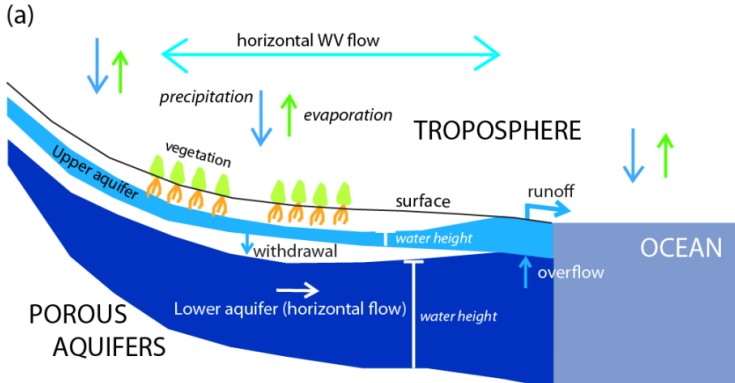

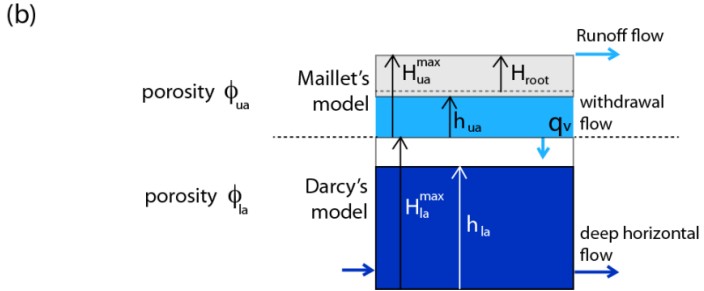


*Figure 4. a) simplified water transfer model including troposphere / vegetation / hydrosphere*
*interactions ; b) equivalent model using notations used throughout the text and eqs. 7 and 8. Case*
*associated to eqs. 7a and 8a is represented.*

Maximum WV concentration in the atmosphere is controlled by the Clausius-Clapeyron (C-C)
relation integrated over the whole tropospheric column, thus providing the upper estimate of precipitable
water vapor (PWV$_{C-C}$). We adapt here the formulation of Reitan (1963):

$PWV_{C-C} = 0.01 \cdot exp\,[0.1102 + 0.06138T]$ (eq. 2)

where $T$ is ground temperature in °C (Fig. 5). For sake of simplicity, we do not consider here the
variation of ground temperature as a function of altitude, meaning that temperature in eq. 2 is
independent on ground elevation. The maximum water vapor concentration $C_{C-C}$ can be expressed as:




$C_{C-C} = \frac{\rho_w g}{P_{atm}} PWV_{C-C}$  (eq. 3)

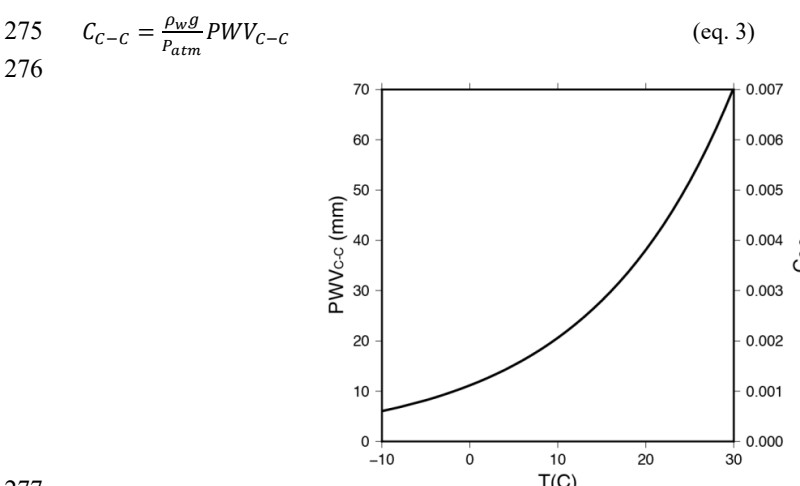

*Figure 5. Maximum precipitable water vapor (PWV$_{C-C}$) and concentration (C$_{C-C}$) associated to the*
*Clausius-Clapeyron law (from Reitan, 1963)*

The precipitation rate largely depends on the relative humidity of the atmosphere $RH$ defined as
$100(C/C_{C-C})$. Also, it has been noticed that the probability of precipitation below a given saturation
threshold is very low, while it is rapidly and non-linearly increasing around this threshold (Neelin et al.,
2009). We mimic this behavior using a power law formulation controlled by its exponent n (ref?) to
explicitly define the precipitation rate (eq. 4). In addition, geochemistry studies provide some values of
water droplet residence time in the atmosphere ranging between 2 and 10 days (Läderach and Sodemann,
2016; Gimeno et al., 2012). Therefore, we choose a simplified relationship where the rain rate is
controlled by a characteristic time $t_{rain}$. Despite the relationship between extreme precipitations and
C-C is not always linear for temperature higher than 15°C (Drobinski et al., 2016), we make the
simplified hypothesis that rain rate $\dot{P}$ scales with the maximum amount of tropospheric water $PWV_{C-C}$,
leading to the following constitutive relation (see also Figure 6):

$\dot{P} = \frac{PWV_{C-C}}{t_{rain}} \left(\frac{RH}{100}\right)^n$  (eq. 4)





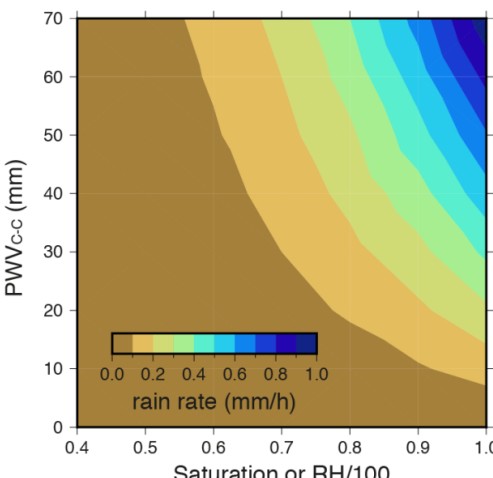

*Figure 6. Rain rate associated to eq. 4 and Table 1*

3.2 Evaporation and evapotranspiration

Due to the complexity of evaporation processes acting over water bodies (lakes and ocean), experimental and theoretical studies do not converge onto a unique formula for the evaporation rate $\dot{E}$. Fortunately, only few atmospheric parameters control evaporation: wind, relative humidity of the lower atmosphere ($RH$), surface temperature and solar radiation (Filimonova and Trubetskova, 2005; Singh et al., 1995). On land, since plant evapotranspiration results from complex physical and biological mechanisms, potential evapotranspiration $ET_0$ appears to be the most common concept defined as the theoretical rate of plant (shallow rooted grassland) transpiration when soil water content is not limiting and ensures optimal stomatal conductance, associated to the Penman-Monteith formulation (Penman, 1948; Monteith, 1965). However, this formula is not easy to use for data assimilation and efforts were made to associate $ET_0$ with atmospheric observations as for the oceanic case, that is relative humidity, surface temperature, and solar radiation. Because water bodies and vegetation share common controlling parameters, we find convenient to use the polynomial formula of Alexandris and Kerkides, (2003) to model within a single equation both oceanic evaporation and empirical plant transpiration:

$$ET_0 = f[S_{irr}, T, RH] \qquad \text{(eq. 5)}$$

where $S_{irr}$ represents solar irradiance and f a polynomial function of degree 2 in $S_{irr}$, T and RH. From an integrative model viewpoint, this assumption presents the interest to embed into a single parametrization vegetation evapotranspiration and water bodies evaporation (Fig. 7). On land, we introduce a scaling coefficient $F_c$ representing the impact of leaf area on kilometer scale transpiration. We therefore mimic a desert using $F_c = 0$ (assuming that no evaporation takes place) and a full forest coverage using $F_c = 1$. The effective potential evapotranspiration formula is given by:

$$ET_p = F_c \cdot ET_0 \qquad \text{(eq. 6)}$$





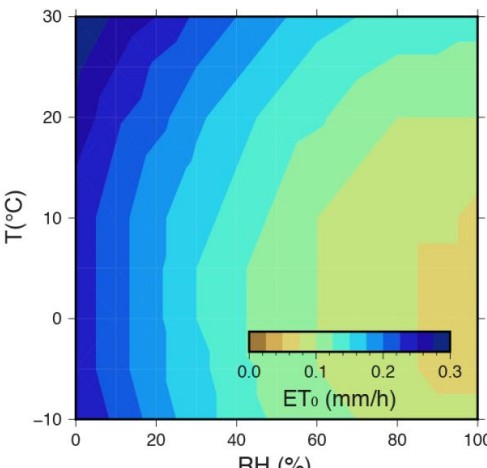

*Figure 7. ET$_0$ as a function of relative humidity RH and temperature T (from* Alexandris and Kerkides,
2003).
Even if potential evapotranspiration may reach several mm per day, real plant evapotranspiration
rate ultimately depends on the soil water amount that roots are able to extract. We consider a model of
trees equipped with roots reaching a depth $H_{root}$, assuming trees living over a shallow aquifer of
thickness $H_{ua}$ in which the water height $h_{ua}$ dynamically evolves between 0 and $H_{ua}$ due to root uptake
and underground water motion (see Fig. 4b). Therefore, real evapotranspiration rate $\dot{E}$ takes two
admissible values:
$h_{ua} \leq H_{ua} - H_{root}$    *then* $\dot{E} = 0$                                    (eq. 7a)
$h_{ua} > H_{ua} - H_{root}$    *then* $\dot{E} = ET_p$                                (eq. 7b)
3.3 Runoff and underground water flow
Rainwater reaching the soil surface follows one of the three paths: (1) shallow infiltration, root
uptake and evapotranspiration, (2) deep infiltration, or (3) surface runoff by direct river connection or
by upper aquifer overflow. We implement upper aquifer overflows using the following conditions for
the model, insuring that $h_{ua} = H_{ua}$ in eq. 8b so that runoff $\dot{R}$ takes place:
$h_{ua} \leq H_{ua}$ then   $\dot{R} = 0$                                        (eq. 8a)
$h_{ua} > H_{ua}$ then   $\dot{R} = d\,h_{ua}/dt$                                (eq. 8b)
Modeling groundwater transfer is a vast research field implying a variety of flow processes and
water/rock interactions (Ntona et al., 2022). One difficulty of setting a continent-scale hydrogeological
model is the high heterogeneity of underground water aquifers due to interplay between geological
processes, relief, topography and surface rivers, thus resulting in a great diversity of groundwater
residence times, concepts and models. As for the troposphere, we assume that water mass conservation
is achieved and we simply distinguish two gravity-controlled water flow processes. The upper aquifer
is treated like a partially saturated zone formed by the soil and subsurface rocks. A useful soil-water
interaction is provided by the Richards equation (Van Dam and Feddes, 2000). However, this model
incorporates many soil parameters that renders it too complex for our purpose. We adopt a simpler model
derived from Maillet's reservoir model (Maillet, 1905; Fleury et al., 2007). This reservoir has a thickness
$H_{ua}$ already defined, and its withdrawal flow $q_v$ at a given horizontal location is provided by the upper
aquifer storage $\phi_{ua} h_{ua}$ ($\phi_{ua}$ being the porosity of the medium) divided by the characteristic time $t_{ua}$ of
the Maillet's reservoir:



$q_v = \frac{\phi_{ua} h_{ua}}{t_{ua}}$                               (eq. 9)
The lower aquifer receives water from above and insure near horizontal flow according to hydraulic
head gradient. We use the original Darcy's formulation (Hubbert, 1956) stating that horizontal flow is
occurring as the product of the hydraulic conductivity $K$ and the hydraulic head gradient of the lower
reservoir free surface.  Estimate of average hydraulic conductivities for large scale aquifers is dependent
on many geological and mechanical parameters. Such an integrated parameter is provided at basin-scale
using geostatistic methods (e.g. El Idrysy and De Smedt, 2007) and we adopt conductivity values
ranging from $10^{-3}$ to $10^{-1}$ m/s. The aquifer hydraulic head involves the topographic elevation $z$ and the
lower aquifer dynamic height $h_{la}$ that are both depending on horizontal x-coordinate. The underground
water advection is controlled by the following conservation equation:
$\frac{\partial h_{la}}{\partial t} = div\left[-K grad_x(z + h_{la})\right] + q_v$           (eq. 10)
Where the upper aquifer withdrawal flow is the source term of the lower aquifer (note the similarity
between eqs. 10 and 1). Water overflows into the upper reservoir if the following condition is met:
$h_{la} \geq H_{la}$                                   (eq. 11)
meaning that $q_v$ becomes negative in order to insure the compatibility equation $h_{la} = H_{la}$. For such a
case, upper aquifer fills from below according to the situation pictured in Fig. 4a (near the coast).
3.4 Numerical formulation
Despite our goal to provide a parsimony formulation of the global water cycle, we are left with a fairly
complex dynamical system including 3 variables ($C$, $h_{ua}$ and $h_{la}$), 11 equations (1-11) and more than
thirty model parameters (Table 1). We therefore implement a dedicated numerical treatment combining
finite elements for horizontal water transfer into the troposphere and the lower aquifer, coupled to a
finite difference scheme for vertical transfer (evaporation and vertical infiltration) in the upper reservoir.
A FORTRAN 95 code was specifically developed to compute dynamic water exchanges between ocean,
troposphere and hydrosphere. We perform different tests to insure solution stability with respect to time
and space discretization, also checking mass conservation over time.
Each experiment corresponds to 40 elements (100km per element) in order to capture solutions
spatial variations, using a 1-hour time step to ensure stability of the finite difference scheme. The
physical duration of each experiment is set to 300 years in order to reach stationary solution for all
presented cases. The computation is achieved after 17s using a 3.4 GHz Intel Core i5 processor (2013).






TABLE 1

MODEL PARAMETERS AND VALUES. Model's variables are given in bold, parameters variations appear in red.

| Symbol | Unit | Used values | Description |
|---|---|---|---|
| GEOMETRY | | | |
| $x$ | m | | Horizontal coordinate |
| $z$ | m | | Topographic elevation |
| $z_{max}$ | m | [100; 1000] | Maximum topographic elevation |
| $L_o$ | km | 2000 | Ocean length |
| $L_c$ | km | 2000 | Continent length |
| TROPOSPHERE | | | |
| **$C$** | **kg/kg** | | **Water vapor concentration** |
| $C_{C-C}$ | kg/kg | | Maximum water vapor concentration |
| $S_{irr}$ | W/m$^2$ | | Solar irradiance |
| $P_{atm}$ | Pa | $10^5$ | Ground atmospheric pressure |
| $T_{ocea}$ | °C | 20 | Oceanic air temperature |
| $T_{cont}$ | °C | [17; 20; 23] | Continental surface temperature |
| $PWV$ | m | | Precipitable water vapor |
| $PWV_{C-C}$ | m | | Maximum precipitable water vapor |
| $\dot{P}$ | m/s | | Precipitation rate |
| $P_c$ | m | | Cumulated precipitation |
| $\rho_w$ | kg/m$^3$ | $10^3$ | Water density |
| $D_h$ | m²/s | [$10^3; 10^4; 10^5$] | Horizontal dispersion |
| $v$ | m/s | [$-1; 0; +1$] | Wind velocity |
| $d_c$ | - | 0.5 | Cloud spatial density |
| $n$ | - | 4 | Saturation exponent |
| $t_{rain}$ | days | 3 | Rain critical time |
| VEGETATION | | | |
| $F_c$ | - | [0; 0.1; 1] | Forest coverage ratio |
| $ET_0$ | mm/day | | Reference evapotranspiration rate |
| $ET_p$ | mm/day | | Potential evapotranspiration rate |
| $H_{root}$ | m | 5 | Rooting depth |
| $\dot{E}$ | m/s | | Real evaporation rate |
| $E_c$ | m | | Cumulated evaporation |
| AQUIFERS | | | |
| **$h_{ua}$** | **m** | | **Upper aquifer water height** |
| $H_{ua}^{max}$ | m | 10 | Upper aquifer thickness |
| $\phi_{ua}$ | - | 0.1 | Upper aquifer porosity |
| $S_{ua}$ | m | | Upper aquifer storage ($\phi_{ua} h_{ua}$) |
| $t_{ua}$ | days | [$10^1; 10^2; 10^3$] | Upper aquifer withdrawal time |
| $q_v$ | m/s | | Upper aquifer withdrawal rate |
| **$h_{la}$** | **m** | | **Lower aquifer water height** |
| $H_{la}^{max}$ | m | 100 | Lower aquifer thickness |
| $\phi_{la}$ | - | 0.02 | Lower aquifer porosity |
| $S_{la}$ | m | | Lower aquifer storage ($\phi_{la} h_{la}$) |
| $K$ | m/s | [$10^{-3}; 10^{-2}; 10^{-1}$] | Lower aquifer hydraulic conductivity |
| $\dot{R}$ | m/s | | Runoff rate |
| $R_c$ | m | | Cumulated runoff |
| $S_{rel}$ | % | | Overall aquifers relative storage |





**4. Results of numerical experiments**

4.1 Geometry, boundary conditions, external forcing and initial state

We aim to understand the physical factors controlling average continental precipitation $P_c$ at a large spatial scale. We therefore design a 2000 km length continent surrounded by two oceans of equivalent sizes (Fig. 8), assuming the following conditions:
- A constant temperature of 20°C is used with tuning solar irradiance for obtaining an oceanic annual precipitation of about 1m (average precipitation rate of 0.12mm/h). We therefore do not account for diurnal and seasonal dynamics.
- Initial conditions correspond to (1) a dry troposphere (C = 0 in eq. 1) and (2) dry continental aquifers ($h_{ua}$ = 0 and $h_{la}$ = 0). Continental evapotranspiration therefore equals zero at the beginning of each experiment, meaning that water transport from the ocean via the troposphere is the sole way to fill upper and lower aquifers;
- Wind velocity can be set to zero to study the dispersion effect of the troposphere.

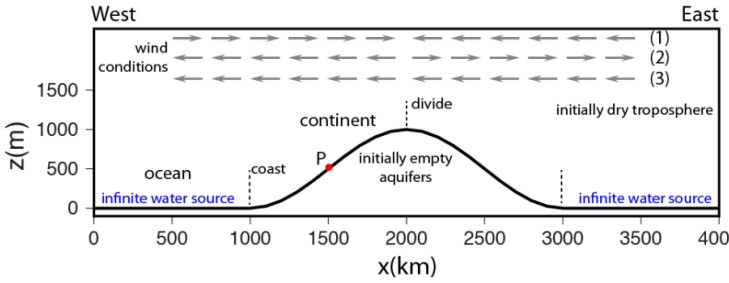

*Figure 8. topography and initial conditions for troposphere and continental aquifers. Solid line figures topography elevation, while grey arrows represent wind conditions used in 4.4 section: (1) convergent system (upwind) from ocean to continent; (2) divergent system (downwind) from continent to ocean, and (3) through wind from east to west. Red dot (P) refers to the time evolution of Fig. 12.*

4.2 Parametric study: vegetation and hydrology

In a first set of 18 experiments, we assume a tropospheric horizontal dispersion of $10^4$ m/s$^2$ (Table 1, see also Pisso et al., 2009). Two values of 0.1 and 1 are assumed for forest coverage ratio in order to mimic respectively sparse and dense forest coverage. We tune upper aquifer withdrawal times over 3 orders of magnitude from 10 days (permeable aquifer) to $10^3$ days (2.7 years, almost impervious medium). We also test the effect of deep aquifer conductivity variations ($10^{-3}$ - $10^{-1}$ m/s).

All experiments display a quasi-invariance of oceanic precipitation despite the variation of continental parameters over 3 orders of magnitude, indicating that continental parameters have a marginal impact on open sea evaporation controlled by vertical transfer between rain rate (eq. 4) and evaporation (eq. 5). We therefore present annual averaged continental precipitation and evapotranspiration normalized by oceanic precipitation ($P_{norm}$ and $E_{norm}$ on Fig. 9).

Sparse forest experiments ($F_c$=0.1) display a modest normalized precipitation on continents never exceeding 20% of the oceanic value over the whole aquifer parameters range (Fig. 9a). By contrast, dense forest experiments ($F_c$=1) reveal a larger precipitation, increasing from 15% (high aquifer conductivity) to 90% (low aquifer conductivity) as shown by Fig. 9b. Evapotranspiration variations are closely associated with precipitation trends for both sparse and dense experiments (Fig. 9c,d) suggesting a causal link between these two quantities.

Considering aquifer's storage (defined as % of the full capacity $S_{ua} + S_{la}$), smaller differences are observed between sparse and dense cases (Fig. 9e, f). Rather, deep aquifer conductivity appears like the main controlling factor with lower values leading to higher water storage. Upper aquifer withdrawal times also display significant impact on water heights, smaller values favoring lower aquifer filling. It



emerges that dense forest water storage is higher than its sparse counterpart for high conductivity and
high withdrawal time, while this storage becomes smaller than sparse water storage for low conductivity
and low withdrawal time. Therefore, water storage appears to be balanced by the whole water cycle
dynamics, including evapotranspiration, aquifer's parameters, and precipitation.

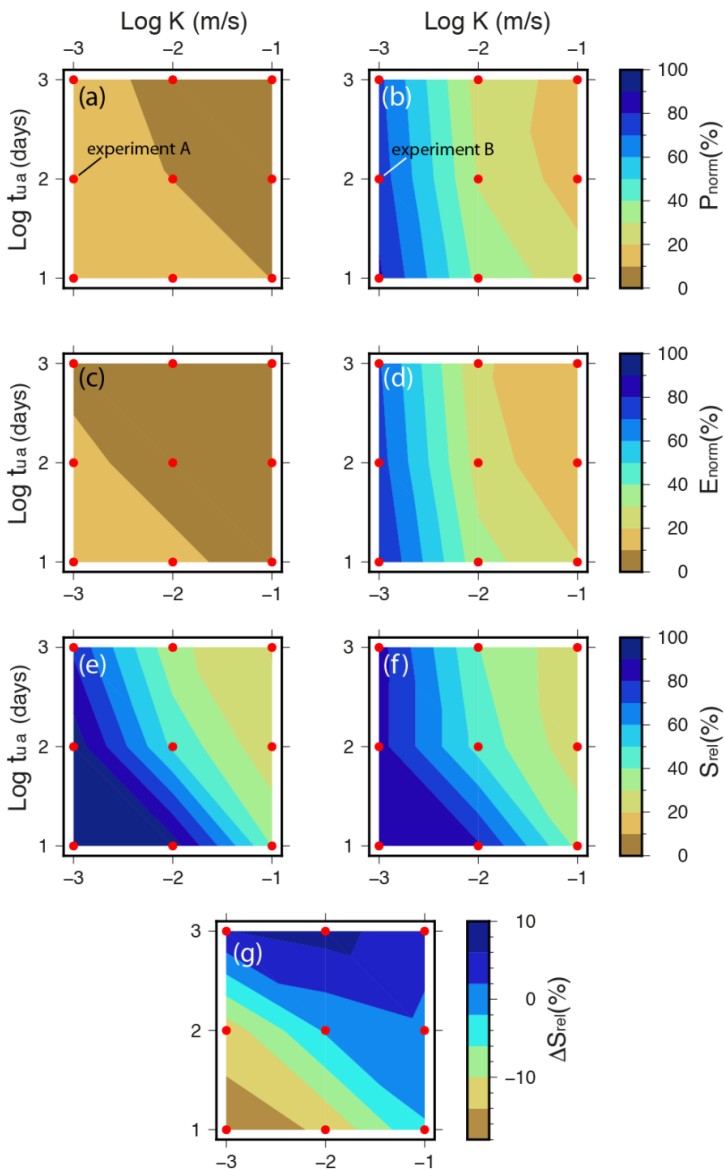

*Figure 9. Annual continental precipitation relative to its oceanic value as function of lower aquifer*
*hydraulic conductivity and upper aquifer withdrawal time after 300yrs; each circle represents an*
*experiment while colored maps result from linear interpolation; (a, b) annual precipitation*
*normalized by oceanic precipitation for respectively sparse and dense forests in %; (c, d) annual*
*evapotranspiration normalized by oceanic precipitation for respectively sparse and dense forests in*
*%; (e, f) overall aquifer storage in % of its full capacity for respectively sparse and dense forests; (g)*
*dense forest storage minus sparse forest storage in %.*




4.3  Sparse and dense forests experiments

To understand the causal link between model's parameters and dynamical outputs such as
evaporation, precipitation and aquifer storage, we track temporal and spatial evolution of some of the
proposed experiments. We present two solutions corresponding to a withdrawal times of 100 days
(permeable aquifer) and an aquifer conductivity of $10^{-3}$ m/s (named experiments A and B on Fig. 9a and
Fig. 9b, respectively). For a sparse forest, the solution displays a pronounced WV concentration gradient
near the coastline, leading to a constant WV concentration for times larger than 200 yrs (steady state
solution, see Fig. 10a). Evapotranspiration and precipitation reach the same constant value ($\dot{E} = \dot{P}$) on
the continent (Fig. 10b), revealing the lack of horizontal WV flow. Near the coastline, aquifers overflow
and produce some local rivers while water transport occurs underground inland (Fig. 10c). Due to low
aquifer conductivity, aquifers are 90% full throughout the continent (Fig. 10d) as also displayed on Fig.
9e.

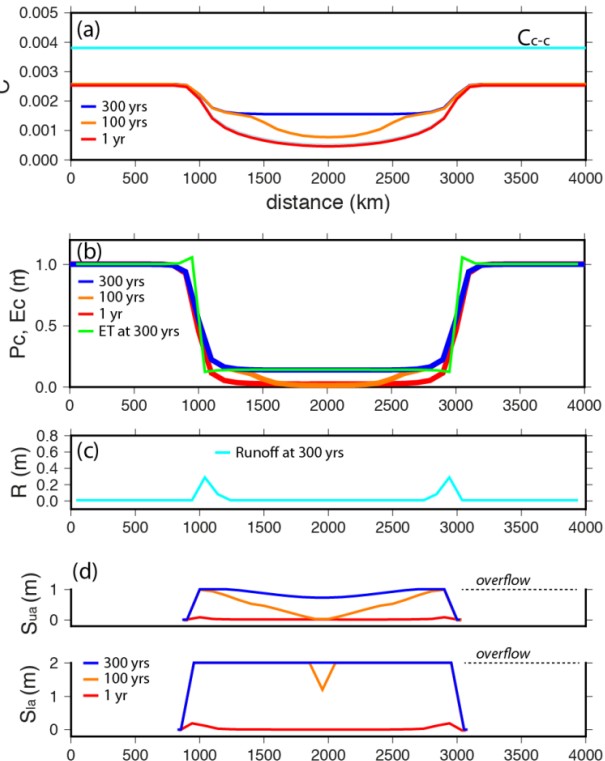

*Figure 10. Steady state solutions (sparse vegetation, experiment A) (a) troposphere WV concentration*
*(red-orange-blues lines) and C-C concentration yield (cyan line); (b) cumulated precipitation (red-*
*orange-blues lines) and evaporation (green); (c) runoff (cyan); (d) upper and lower aquifer storage.*
Dense forest solution reveals a different evolution with a progressive WV concentration increase
during 300 years until it reaches a much higher value than sparse value experiment (Fig. 11a).
Evapotranspiration and precipitation also become equal for times larger than 200 years (Fig. 11b). Due
to active upper aquifer pumping by the trees, no runoff takes place (Fig. 11c) as the upper aquifer is





filled at only 50% of its maximum capacity (Fig. 11d). As a result, dense forest overall aquifers are only
partially filled and depleted by 15% with respect to sparse forest (Fig. 9g).

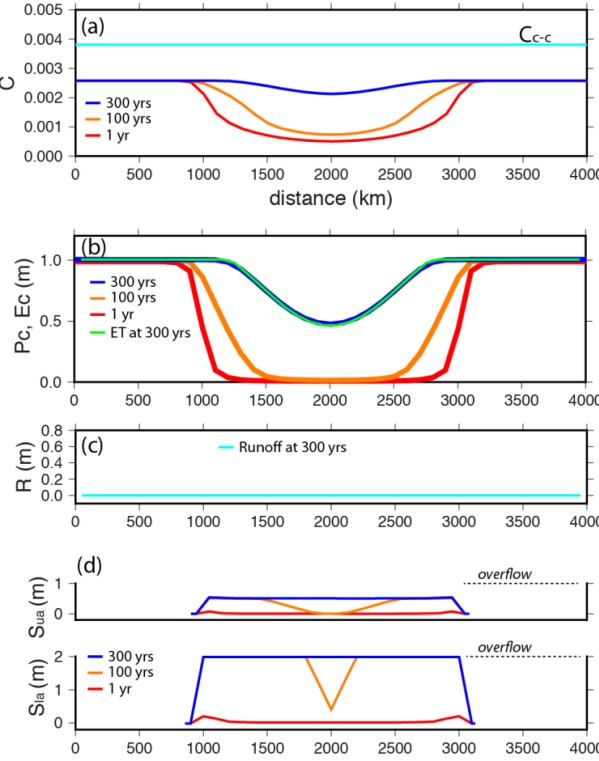

*Figure 11. Steady state solutions (dense vegetation, experiment B) (a) troposphere WV concentration*
*and C-C concentration yield (cyan line); (b) cumulated precipitation and evaporation; (c) runoff; (d)*
*upper and lower aquifer storage.*
In order to identify functional interplays between atmospheric, biophysical and hydrological
solutions, we track time evolution of dense forest experiment (B) at a point P located 500 km from the
coast (x=1500 km, Fig. 8 and 11). A staggered evolution takes place from empty reservoirs to steady-
state values following atmospheric and hydrologic steps:
(1)  initial WV concentration (set to 0) rapidly increases within 1 year, resulting into a step in Fig.
12a due to horizontal WV dispersion from ocean to continent;
(2)  progressive deep aquifer filling from 0 to 40 years;
(3)  then, deep aquifer overflows into the shallow aquifer that progressively fills from 40 to 130
513            years;
(4)  upper aquifer head reaches root depth at 130 years thus allowing evapotranspiration increase
until its stabilized value at 145 years. Due to EVT increase, WV concentration positively
jumps, thus triggering precipitation by a factor of 7.5 (from 0.1m/yr at 130 years to 0.75 m/yr
at 145 years).



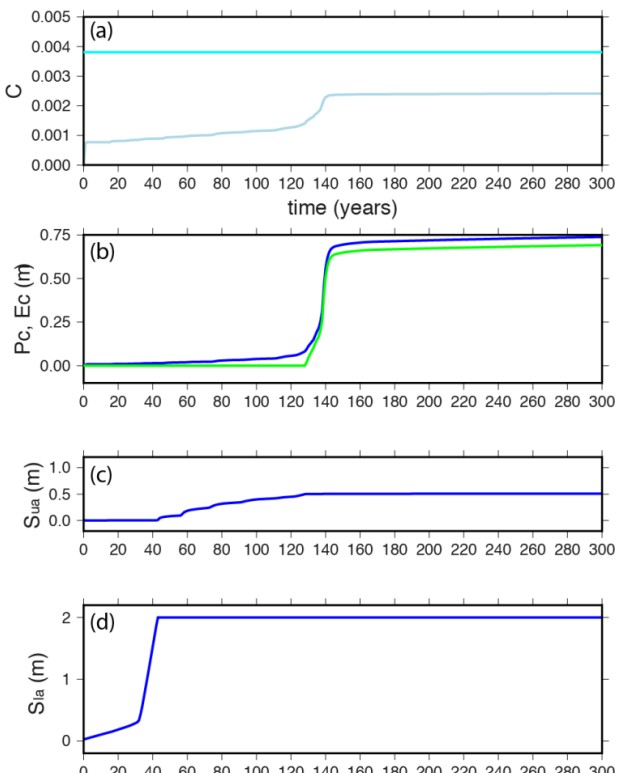


*Figure 12. Time evolution at 500km from coast (dense vegetation, experiment B) (a) troposphere WV*
*concentration and C-C concentration yield (cyan line); (b) cumulated precipitation (blue line) and*
*evaporation (green line); (c) upper aquifer storage; (d) lower aquifer storage.*

This retroaction chain involving atmospheric, biophysical and hydrological components
systematically occurs in all our models with dense forest parameters, the final consequence being an
increase of continental precipitation up to a stabilized value (Fig. 9 and 12). The magnitude of the
precipitation increase depends on the capacity of the vegetation to deliver WV to the atmosphere.
However, complexity emerges due to the coupling between evapotranspiration capacity and the other
elements of the water cycle and especially the groundwater deep aquifer. We therefore explore how
tropospheric, topographic, geological and biophysical parameters influence steady state continental
precipitations, using the same range of vegetation coverage (0.1 and 1) and lower aquifer conductivity
($10^{-3}$ - $10^{-1}$ m/s) as for the preceding experiments.
4.4 Tropospheric parameters: dispersion, temperature and wind
Because long range chemical dispersion $D_h$ is expected to largely vary with atmospheric
conditions, we explore the influence of respectively low ($10^3$ m/s$^2$) and high dispersion values ($10^5$ m/s$^2$)
on normalized precipitation with respect to our reference value of $10^4$ m/s$^2$ (corresponding to red curves
on Fig. 13). Sparse vegetation cases reveal a significant precipitation increase (20-100%) for high
dispersion values (Fig. 13a) and no change for low dispersion. Dense vegetation experiments magnify
this trend with a 30-120% precipitation increase for high dispersion values (Fig. 13b). Noticeably, the
largest precipitation variation occurs for low hydraulic conductivity ($10^{-3}$ m/s), while variations remain
modest for high hydraulic conductivity ($10^{-1}$ m/s).





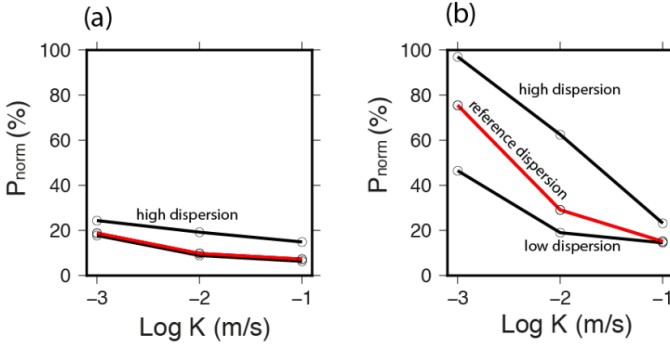

*Figure 13. Normalized precipitation as a function of tropospheric dispersion (low=$10^3$ $m^2/s$; high=$10^5$*
*$m^2/s$, black lines; reference=$10^4$ $m^2/s$, red lines) for different hydraulic conductivities (a) sparse*
*vegetation (b) dense vegetation.*
We then vary continental temperature with respect to the reference value of 20°C with a -3°C
decrease and +3°C increase, the oceanic temperature remaining at 20°C. The impact of those changes is
negligible for sparse vegetation cases (Fig. 14a). It appears that the largest changes occur for dense
vegetation coupled to low hydraulic conductivity (Fig. 14b). Lower temperature increases precipitation
(because the maximum precipitable water vapor PWV$_{C-C}$ limit decreases, see Fig. 5). Higher temperature
shows the opposite effect and the precipitation drops from 78 % to 58 %.

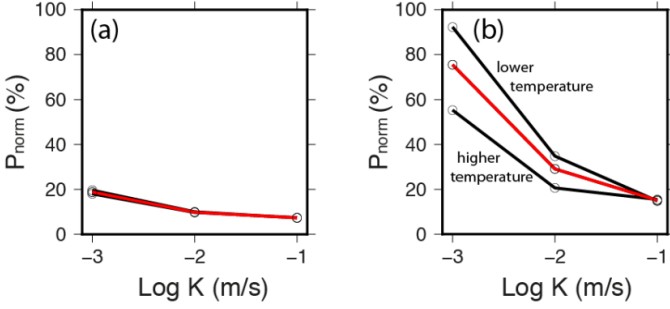

*Figure 14. Normalized precipitation as a function of continental temperature (lower=17°C;*
*higher=23°C) for different hydraulic conductivities (a) sparse vegetation (b) dense vegetation.*
All previous experiments stand for no advection in eq. 1 (wind velocity $v$ set to zero). We test
here three wind conditions: (1) a fully convergent system (upwind) of 1 m/s from ocean to continent,
(2) a fully divergent system (downwind) of 1 m/s from continent to ocean, and (3) a through wind of 1
m/s combining upwind on the right (east) side of the model and downwind on the left (west) side of the
model (see also Fig. 8 for wind conventions). Imposing a wind condition dramatically changes
precipitation patterns for both sparse (Fig. 15a) and dense (Fig. 15b) cases. Downwind condition leads
in any case to zero precipitation, suggesting that some continental regions may become completely dry
if this wind regime is permanent. At the opposite, upwind cases maximize precipitation by a ratio,
relatively to the zero-wind cases, ranging from 2.2 to 7. Interestingly, dense forest cases reveal that
continental precipitation may exceed the oceanic value due to a positive combination of dispersion,
advection and evapotranspiration bringing moisture to the continent.





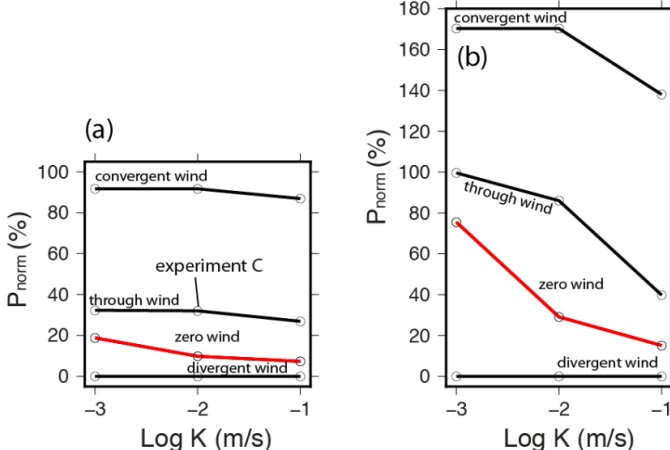

*Figure 15. Normalized precipitation as a function of wind condition as described in the main text (a)*
*sparse vegetation (b) dense vegetation.*

Through wind condition is insightful as it represents upwind/downwind combination often occurring on
Earth with trade winds (tropical zone) and westerlies (tempered zone). Although such experiments
present a zero net wind budget for the continent (null divergence in eq. 1), large precipitation increases
(50% - 150%) are observed for both sparse and dense vegetation cases with respect to zero wind cases.
To highlight this point, we present in Fig. 16 through wind spatial patterns for sparse vegetation case.
Upcoming wind from the east leads to asymmetrical WV concentration (Fig. 16a) with higher
concentration upwind. Accordingly, precipitation is higher on east coast (Fig. 16b) while west coast
remains extremely dry including its oceanic part. Due to low evapotranspiration and low aquifer
hydraulic conductivity, aquifers are almost full across the whole continent (Fig. 16c) while runoff mainly
takes place close to upwind coast to the east (Fig. 16d).



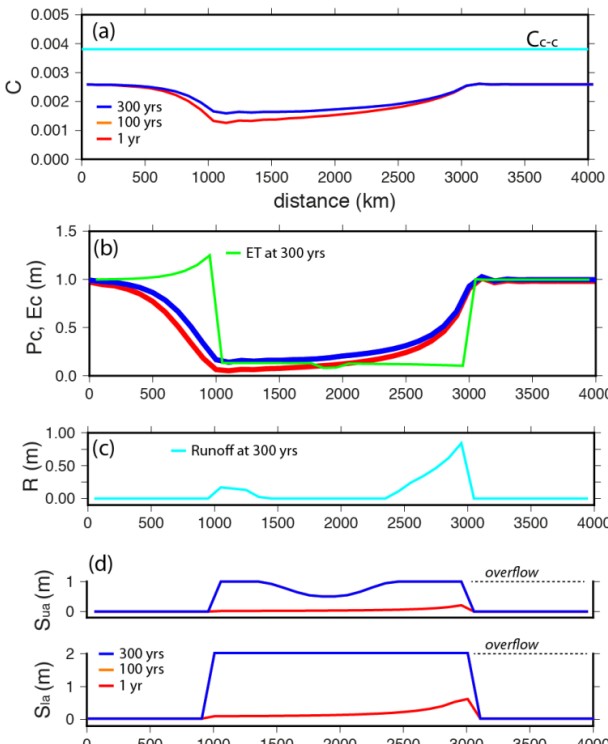

*Figure 16. Steady state solutions (sparse vegetation, through wind condition, experiment C on Fig. 15a (a) troposphere WV concentration; (b) cumulated precipitation and evaporation; (c) runoff; (d) upper and lower aquifer storage.*

### 4.5 Topographic parameters

Topography influences both orographic conditions (modeled here through temperature and integrated WV) and hydrological flow. We recall that surface temperature in eq. 2 (Reitan formula) is independent on ground elevation. Therefore, topography only modifies hydrological flow and the orographic effect on precipitation is not accounted. We check hydrological influence by introducing a low topography with a maximum elevation of 100 m, 10 times smaller than for the previous models. In this way, we modify the contribution of the hydraulic head of underground water motion (eq. 10). While this parameter change has a relatively minor influence for sparse vegetation cases (Fig. 17a), precipitation increases up to 150% for dense vegetation cases depending on lower aquifer conductivity values (Fig. 17b). This water cycle enhancement is due to a global interplay between horizontal water flow lowering, reservoir filling and the increase of evapotranspiration as already detailed above (see Fig. 12).





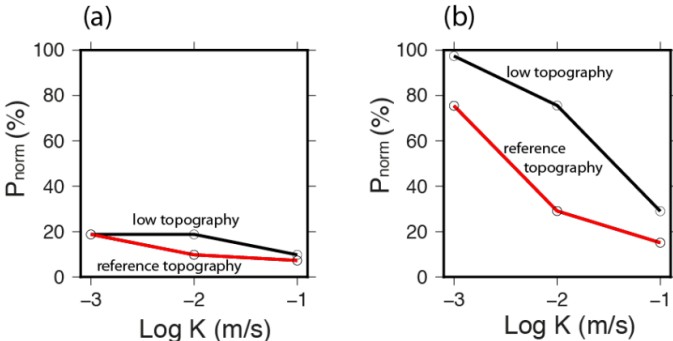

*Figure 17. Normalized precipitation as a function of topography condition (red line=reference, black*
*line=low topography) (a) sparse vegetation (b) dense vegetation.*

4.6 Forest geographic location

Until now we have considered a homogeneous vegetation density with sparse ($F_c = 0.1$) and
dense cases ($F_c = 1$) cases. We now consider a 400 km-long densely forested area ($F_c = 1$) at a distance
to the coast varying from 0 km (coastal forest) to 800 km (central forest), the rest of the continent being
fully deforested ($F_c = 0$). We also compute, for reference, a case without any forest ($F_c = 0$). The goal
is to evaluate how tropospheric WV transport and evapotranspiration alter precipitation distribution for
these two distinct environmental settings shown in Fig. 18a. We compute the solution at t=300 yrs for a
tropospheric dispersion of $10^4$ m$^2$/s and no wind, thus providing equivalent normalized precipitation as
a function of coast-to-forest distance. The coastal forest case (Fig. 18b) reveals an efficient water transfer
from the ocean to the continent. Indeed, significant precipitation amounts are observed as far as 400 km
inside the continent due to the forested zone providing high evapotranspiration, the annual precipitation
of the coastal area being in the range 0.8-1m, dramatically larger than the one associated to a deforested
area (like the right coast of the model receiving only 0.05-0.1m). Although central forest is set to the
same spatial extension than coastal forest, the precipitation feed-back appears to be much weaker with
an average precipitation value of 0.2 m over the central forest (Fig. 18c).



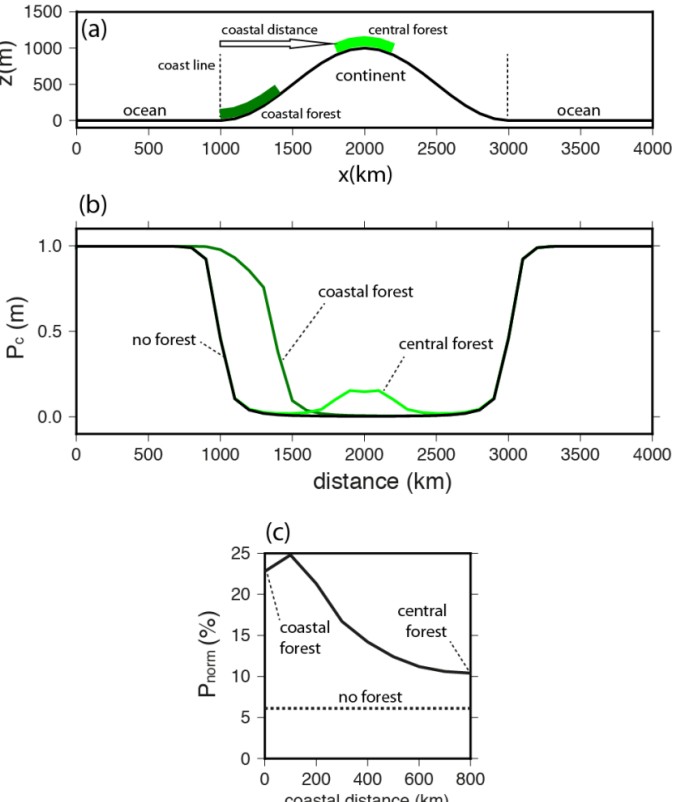

*Figure 18. (a) Location of respectively coastal and central forest cases (b) spatial precipitation*
*distribution for no forest (black line), coastal forest (dark green curve) and central forest (light green*
*curve) (c) average continental precipitation normalized by its oceanic counterpart as a function of*
*coast-to-forest distance (solid curve) and for a non-forested area (dashed line)*
**5. Discussion**
5.1 Parsimony model analysis
Water cycle experiments presented in the previous section embed four interconnected factors: (1)
horizontal water vapor flow (2) liquid water production by condensation (3) horizontal groundwater
flow (4) evapotranspiration by the vegetation. The idea that continental precipitation emerges as a
combination of these dynamical processes is not new (Foley et al., 2003), but its quantitative exploration
was until now restricted to the use of coupled atmospheric, hydrologic and vegetation models involving
a high complexity level, thus rendering unreachable a user-friendly research. We mean by user-friendly
that testing alternate constitutive equations, special boundary conditions and more generally new
working hypothesis should be easy to perform in order to foster new discoveries. State-of-the-art
meteorological codes, operable by specialized users on supercomputers, do not match this requirement.
By contrast, our parsimony model delivers fast-track continental precipitation simulations,
opening doors to hundreds of ad-hoc models. This strategy allowed us to test numerous parameters
associated to:
- atmosphere (horizontal tropospheric dispersion, wind velocity, continental surface temperature);
- biosphere (forest coverage ratio, potential evapotranspiration or ET);





- lithosphere (topographic elevation, upper aquifer withdrawal time, lower aquifer hydraulic
conductivity).
Starting from empty atmosphere and underground water reservoirs, the three independent variables of
the model (WV concentration $C(x)$, upper and lower aquifer water height $h_{ua}(x)$ and $h_{la}(x)$, see table
1) dynamically evolve towards a steady state solution, meaning that incoming oceanic WV becomes
balanced by water exported by continental aquifers and surface runoff (Fig. 4).
Parameter's space exploration leads to the following propositions:
a. when models embed limited biophysical activity (sparse forest and/or low potential ET), coastal
areas emerge as a damping zone for oceanic precipitation, the inner continent remaining dry.
This situation is clearly associated to a decay of WV concentration with a length scale depending
on the tropospheric dispersion. This result aligns with previous work (Ogino et al., 2017;
Yamanaka et al., 2018) describing the ocean/continent limit as a "coastal dehydrator", a concept
also illustrated by the relative humidity map of Australia (Fig. 2c).
b. when dense forests and therefore high potential ET take place, precipitation extends inland if
the upper aquifer is able to fill. In our model, this implies that the lower aquifer can accumulate
enough water to overflow into upper aquifer. This condition initiates biophysical (forest) water
pumping and evapotranspiration, adding humidity to the atmosphere, resulting into dramatic
rainfall increase even in remote inland regions (Fig. 11a,b) compared to the case with sparse
forest (Fig. 10a,b). This result clearly supports MG2007 claim, stating that large forested areas
are the ultimate cause for large and homogeneous continental precipitation. However, the
underlying physical process for such a precipitation pattern is not the one invoked in MG2007
(in short: a wind induced by WV condensation over the forest). Rather, we rally a classical
meteorological view postulating that rain occurs by combining two processes: (1) forest
evapotranspiration that enriches tropospheric WV concentration (vertical flow) and (2)
tropospheric dispersion bringing oceanic moisture at large distance from its production area (3)
opposite groundwater fluxes balanced tropospheric WV dispersion. Starting from empty
aquifers, dynamic evolution take place over tens or hundreds of years thanks to regressive –
initiated along the coast – aquifers filling (Fig. 12).
c. The contrasted rainfall distribution of these two situations (ie. function of forest coverage) above
is modulated by parameters related to topographic/atmospheric/hydrogeologic situations, also
including vegetation dynamics. Among these governing factors, topography and aquifer's
dynamics play a key role. Indeed, lowlands – as they exist in large tropical and boreal forests –
decrease underground Darcy's flow, thus favoring water accumulation in the continent, in turn
enhancing effective ET and WV concentration, finally triggering precipitation (Fig. 17b). Wind
is another factor that directly influences WV coming from oceanic domain in addition to
concentration gradient (see eq. 1). Through wind continent breaks the symmetry of
concentration, precipitation and runoff functions with respect to mid-continent divide. As
observed in natural situations (see the Australian case of Fig. 2), upwind coasts then receive
more precipitation that their downwind counterpart (Fig. 16), due to a combination of WV
concentration asymmetry and orographic effect (which is not modelled here, see comment about
eq. 2).
5.2 Tropospheric dispersion length scale
The precipitation decay from ocean to continent (see Fig. 10b) can directly be linked to model's
parameters if atmospheric equations are condensed into a single expression. Such a compact formulation
can be written when considering the steady state form of eq. 1 and a linear form of eq. 4 ($n = 1$). Then,
equations 1, 3 and 4 provide a second order differential equation with constant coefficients:
$D_h \frac{d^2C}{dx^2} - v\frac{dC}{dx} + \frac{1}{t_{rain}}C = \frac{\rho_w g}{P_{atm}}\dot{E}$   (eq. 12)




$\dot{E}$ being equal to zero and 1m/yr over oceanic and continental domains, respectively. Due to the linearity
of eq. 4, solutions of $C(x)$, $\dot{P}(x)$ and cumulated precipitation $P_c(x)$ are homothetic and can be
explicitly written as the sum of two exponential functions (see Appendix A for demonstration):

$$C(x) = c_1 e^{\lambda_1 x} + c_2 e^{\lambda_2 x} + \frac{\rho_w g}{P_{atm}} t_{rain} \dot{E} \qquad\qquad \text{(eq. 13)}$$

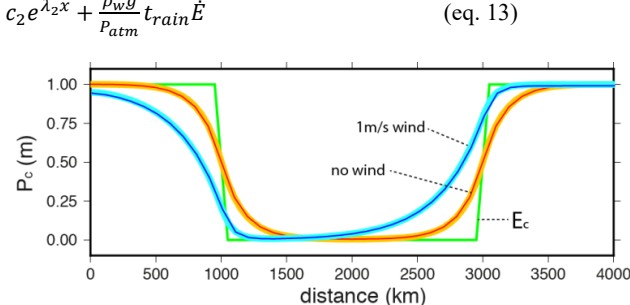

*Figure 19. Spatial precipitation distribution associated to eq. 12. Green curve represents evaporation*
*profile (right hand side of eq. 12). Bold curves (orange= no wind; cyan= -1m/s wind) are associated*
*to analytical solutions (eq. 13) while thin curves (red=no wind; blue=-1m/s wind) correspond to*
*numerical solutions after 100yrs.*

We compute this function (eq. 13) using an evaporation $E_c$ of 1m on the ocean, a null
evapotranspiration on the continent, a dispersion $D_h$ of $10^5$ m²/s, a time scale $t_{rain}$ of 3 days and two
wind values: $v = 0$ and a wind coming from east at 1 m/s. Comparing analytical and numerical solutions
(Fig. 20) lead to an excellent agreement for $n = 1$. In the case of a null wind, $\lambda_1 = \lambda_2$ and eq. 13
provides a simple expression of the characteristic damping distance of the solution:

$$\sigma = 1/\lambda_1 = \sqrt{D_h t_{rain}} \qquad\qquad \text{(eq. 14)}$$

Using modeling value of $t_{rain}$ (3 days, see Table 1), damping distances of 16, 50 and 161 km are found
for respectively atmospheric dispersion values of $10^3$, $10^4$ and $10^5$ m²/s. The above relationships (eq. 13
and 14, fig. 20) may support the conclusion of Ogino et al (2017) concerning dehydration processes with
a characteristic wavelength of 150 km. Indeed, these authors state that "*the dehydration process robustly*
*exists as a global mean feature over a range of a few hundred kilometers from the coastline*". Therefore,
this dehydration length may correspond to the damping distance definition given by eq. 14 (null average
wind) corresponding to $E > P_c$ over oceanic domain while to $E < P_c$ on continental areas.

5.3 Forest physiology evolution and continental colonization

We implemented a formulation allowing to convert potential evapotranspiration rate ($ET_p$) to real
evapotranspiration rate $\dot{E}$, this latter quantity being tuned by available underground water for root uptake
(eq 7a,b). However, we did not consider other physiological factors controlling biophysical forest
dynamics. Three of them matter as far as climate change is concerned:
The first one is associated to the proposition that "*actual rooting depths reflect ecologically*
*optimized responses to weather and climate variability*" (Feddes et al., 2001), as reviewed by Bachofen
et al. 2024. For example, Mediterranean tree species develop deep root systems (more than 10 m depth)
to survive during summer droughts with a root/shoot ratio increase toward drier conditions (Cabon et
al., 2018), following the eco hydrological equilibrium theory (Eagleson, 1982). Following this theory,
leaf area index, driving transpiration, also adjusts to dry conditions (Hoff and Rambal, 2003).
The second one is that forests are self-organized, species diversity promoting global forest
productivity (Morin et al, 2025). These two factors play at time scales associated to individual trees
lifetime and also the time needed to obtain a mature forest, meaning hundreds of years depending on
climate conditions. This suggests that forests are optimized to uptake water from shallow aquifers, both



individually and collectively, with complementary strategies between species leveraging the trade-off
between root depth and drought tolerance (Brum et al. 2019). Because these two factors are likely to
influence evapotranspiration amplitude and seasonality, large forests should modify climate through
physical processes that we have described and modelled.
In addition, a very long-term factor may influence climate and precipitation distribution at
geological time scales: continental life evolution started from small plants species to evolve towards
large trees. We know from geological records that the first episode of forest spreading occurred at
Devonian times (420 to 360 My bp) along with a massive transition of plant physiology, climate and
hydrology Gess and Berry (2024). Quoting from these authors, "*Devonian was a time of significant*
*evolution of plant life, with the establishment of forest stature vegetation for the first time in Earth*
*history. This development is believed to have had massive permanent repercussions; enhancing the*
*chemical weathering of rocks, changing the nature of fluvial systems…*". Among various emerging
physiological factors, the dramatic increase of both plant size and root system is of specific interest to
understand the link between plant evolution and climate change (Fan et al., 2017; Stein et al., 2020;
Laughlin et al., 2021).
To build a retroactive mechanism between forest development and precipitation distribution, we
consider that plant development physiological factors are solar powered. First, a large part of this
radiative power is absorbed or re-emitted as heat flux in infrared energy bandwidth, the thermoregulation
being achieved by latent heat release by plant transpiration (Zeng & Zhang, 2020). This process is partly
modeled here (plant transpiration) by using solar radiation as a parameter (eq. 4). Secondly, a small part
of solar energy is used to create organic matter by photosynthesis. It would be useful to model this
second process in the framework of our model and to setup a plant growth function to complement
evapotranspiration function. Such a coupling between forest dynamics and global water cycle may lead
to the following 3-phase scenario:

a.   Devonian time shelfs (Laurassia and Gondwana) presented a large extension similar to those
of our modern continents. Prior to forest development, plates interiors were likely to be free
of massive evapotranspiration sources. In such a case, our models indicate that large
precipitation should have been limited to coastal areas, remaining inland zones dry.
b.   Triggered by coastal rain, trees growth and initial forest development should have therefore
started along the coast, species equipped by deep root systems being favored in order to
penetrate inside continents. Looking to Fig. 3 of Gess and Berry (2020), *Archaeopteris* paleo-
trees are found in such paleogeographic position. During this initial spreading, precipitation
patterns of forests invading coastal areas should resemble to the transition from black to dark
green curve of Fig. 18b. This suggests that rainfall migrates inland as forests growth and
aquifers fill.
c.   As forest colonization goes on, its rate is modulated by some factors that we modeled: wind,
topography and temperature, and others like soil weathering rate, plant growing rate,
mortality rate and other geological factors. Depending on their influence, forest may have
spread partially or totally over continents due to the positive combination of four factors: WV
horizontal transfer, forest evaporative function, precipitation enhancement and aquifers
filling. This interplay may have induced major climatic shifts during Earth evolution, the
fundamental pattern being that forest colonization may extend inland by converting dry
shelves interiors to humid and rainy continents (see Fig. 11b).

5.4 Holocene climate, forests and civilization fate

The end of Mediterranean Holocene humid period around 6500 years ago is generally attributed
to an astronomical forcing inducing monsoon migration (e.g. Marzin et Braconnot, 2009). However,
these interpretations remain speculative because climate model's predictions are not backed by direct
paleo-meteorological observations. Because Mediterranean and African deforestation took place during
Holocene times, it has also been suggested that a climate modification by human societies might also
have occurred, thus hasting the end of African humid period (Wright, 2017). As an alternate view to
astronomical forcing, our model supports the proposition that human societies may have lowered





precipitation by massive forest clearance, thus depleting aquifers. This hypothesis is supported by the fact that time scales associated to natural forest growth are of hundreds of years, implying that present-day databases are not long enough to display this slow balance evolution. Despite youngly afforested zones may not display obvious meteorological or hydrologic changes (Douville et al., 2024), some studies suggest that this effect is already detectable (e.g. Li et al., 2021). The second element supporting our hypothesis is that deforestation is systematically associated to rainfall decrease and droughts at any latitude and climate (Perugini et al., 2017; Smith et al., 2023).

Contrary to fossil fuel extraction, man induced deforestation started a long time ago but this tendency accelerates at the dawn of Holocene (Williams, 2003) because of two novel factors: animal domestication and cultivated plants spreading. Both activities exerted a direct pressure on forests that have been converted to pasture and grain fields. Some retroactive effects of these changes are well known: food productivity increase, soil erosion and biodiversity drop. Since these effects are visible in many ancient societies since 10 000 years BC, their climatic and hydrologic consequences may have challenged their survival.

Although social and political causes are *in-fine* responsible for instabilities and tragic fates of civilizations, we propose that a physical factor also undermines mankind stability during Holocene times: precipitation decay and droughts due to forest conversion to pasture and cereals fields. To understand how this may occur, we start from one of our model's output: continental rain is tuned by two main parameters: moisture brought by dispersion and oceanic wind and local evapotranspiration. We also need to considerate processes associated to vegetation that are seasonal and largely tuned by temperature and plant physiology. Therefore, grain fields display large evapotranspiration during their growing time and then decline in summer. More specifically, harvest abruptly ends evapotranspiration, inducing sharp decrease of atmospheric humidity. This process is occurring also for plants reproducing by seeds with summer drying up.

Despite we did not explicitly model these seasonal variations, further use of our model may help exploring the relation between ending of evapotranspiration and summer precipitation decay, leading to progressive aquifer depletion. Repeated over decades, long and dry summers may have shifted the time of dry period to middle or early spring, therefore increasing the probability of crop failures, adding starvation to rain shortage. The negative impact of natural forests to pasture or culture conversion on precipitation may also sums up with ground level temperature increase induced by evapotranspiration drop (latent heat effect). We propose that replacing forest by crops may have drastically decrease annual precipitation thus favoring societal disorders. Because water resources and associated cultural practices were vital for ancient Mediterranean societies, decays of some prehistoric societies (Vidale et al., 2018) and the collapse of Bronze age (Vaezi et al., 2022; Cline, 2021) might have been caused by man-induced climate change.

5.5 Implications for land and forest management

The lack of a general acknowledgment of the forest - precipitation retroaction has multiple origins, one of them being the belief that moisture advection by wind stands as the most efficient way to transport oceanic moisture over continent. This interpretation is certainly true for areas presenting a polarized annual wind budget like coasts facing trade winds or westerlies. In such a case, it is likely that forests should cause minor change to precipitation budget and vegetation must adapt to the meteorological forcing.

The situation appears to be different when wind budget is close to zero, implying that dispersion effect plays a key role: moisture gradient invariably transfers WV from a wet ocean to a drier continent (see eq. 1). In such a case, forest evapotranspiration may stimulate precipitation in two ways: (1) bringing WV concentration to its condensation limit, and (2) decreasing lower troposphere temperature by evapotranspiration. We also have seen that specific geographic configurations – like coastal forests – maximize inland precipitation (Fig. 18). Because Neolithic and historical migrations preferentially followed coastal routes all around the world, coastal areas have been more or less constantly deforested. As a result, long term man induced deforestation – for example in the Mediterranean basin – may have severely altered regional precipitation regimes. We may therefore conjecture that the natural equilibrium state of large geographic zones (ie, forest extension and precipitation that would have occurred without man interaction) is completely unknown.





Completing existing works on the beneficial ecological role of forests (Singh et al., 2024; Bastin
et al., 2019), our model brings theoretical support to the conjecture that forests enhance precipitation
(Hoek et al., 2022, Makarieva et al., 2023, Yang et al., 2023). We raise an important factor that may
have been underestimated in the past: the magnitude of the retroaction fully depends on the hydrological
part of the global water cycle system. Indeed, it is well known that forests are pumping into aquifers,
thus increasing the risk of drying sources and decreasing river flows (Zhang et al, 2008) if precipitation
remains constant. But we have seen that this latter assumption is flawed: atmospheric WV transport,
vegetation activity and aquifer dynamics are fully coupled. This physical entanglement implies that three
scientific communities – dynamic meteorology – forest ecology – hydrology – must work together to
define new climatic scenarios. This occurs at a special time when climate projections predict
precipitation decrease, aquifers depletion and drought increase over many continental areas (Tramblay
et al., 2020). This scenario appears to be paradoxical if we consider the physical origin of precipitation:
temperature increase means more sea evaporation, thus more rain available over neighboring countries
if moisture is transported there and condensates. Such a warming/greening/wetting scenario has been
recently revealed by paleobotanists for a hot Miocene period in South America (Ochoa et al., 2025). Our
modeling results indicate that such evolution could be systemic, and that natural afforestation should be
an efficient and ecological way to reactivate precipitation around large evaporative bodies.
**6. Conclusion**

The conjecture that "*forests favor continental scale pluviometry*" received various appreciations
along historic periods. Early observers (Christopher Colombus back to sixteenth century, many eighteen
century engineers and foresters) acknowledged that forests trigger pluviometry thanks to *de visu*
observations, while modern science mostly ignores or disagrees with this opinion. It is interesting to
notice that the revival of this proposition by Makarieva and Gorshkov (2007) – stating that forests induce
precipitation – was either received positively or negatively depending on scientific community. Most of
positive comments came from forest ecology community (eg. Sheil & Murdiyarso, 2009), while
hydrology community considered this conjecture as not outrageous (Wierik et al., 2021), probably
because the impact of deforestation on rainfall decay is clearly demonstrated by data. By contrast,
meteorology and climatology literature did not explore this conjecture, assuming that the underlying
physical explanation of MG2007 is flawed (Meesters et al., 2009).
Through this work, we have shown that the proposition "*forests favor continental scale
pluviometry*" has far reaching digitations: basic meteorological equations, forest physiology,
root/aquifer interactions at different scales, underground water flow. Its overall formulation is so
complex that its understanding is unbearable by reductionist scientific practices that have been
developed during several decades and that form the overwhelming majority of modern scientific papers.
A proper formulation – tractable and understandable – requires a double move: (1) a holistic shift, to
consider together in a unified physical formalism all aspects of the water cycle beyond the ideal view of
Fig. 3, and (2) a parsimony strategy, to define an adequate simplification of the problem, facing the need
to remain in contact with observations. In our attempt to tackle the "biotic pump" problem defined by
MG2007, we have formulated a parsimony model that explain that "*forests favor continental scale
pluviometry*" under certain conditions. Considering that the physical mechanisms of the retroaction have
been presented for the first time in the present paper, we hope that this work will induce further research
with the practical goal of avoiding a dry future for many populations.
**7. Appendix A**
We consider the equation:
$$D\ C'' - vC' - \alpha C = A \qquad\qquad (A1)$$
With $\alpha > 0$, with a computing domain ranging from $0\ to\ 4L$. The solution is split into 3 branches: $C_1$
for x < L, $C_2$ for L<x<3L, C3 for 3L<x<4L. The source A is constant over each of these domains. We
apply homogeneous Neumann conditions at the boundaries for x=0 and x=4L:



$$C_1'(0) = 0 \qquad , \qquad C_3'(4L) = 0 \qquad\qquad (A2)$$

And we also impose the continuity of the solution and its derivatives to ocean-continent transitions:

$$C_1\ (L) = C_2\ (L) \qquad , \qquad C_1'(L) = C_2'(L) \qquad (A3)$$
$$C_2\ (3L) = C_3\ (3L) \qquad , \qquad C_2'(3L) = C_3'(3L) \qquad (A4)$$

We consider a negative wind ($v < 0$) and we choose *a-posteriori* the solution that corresponds to a null derivative for x=0 and x=4L, with setting value of C for x=4L (incoming moisture flow). The characteristic equation of eq. A1 is given by:

$$D\ \lambda^2 - v\lambda\ - \alpha = 0 \qquad\qquad (A5)$$

that corresponds to the following real solutions:

$$\lambda_{1,2} = \frac{v \pm \sqrt{v^2 + 4D\alpha}}{2D} \qquad\qquad (A6)$$

The general solution is given by:

$$C(x) = c_1 e^{\lambda_1 x} + c_2 e^{\lambda_2 x} - A/\alpha \qquad\qquad (A7)$$

The constants $c_1$ and $c_2$ being computed for each region using boundary conditions and ocean-continent transitions values by solving a 6x6 linear system.

## 8. Code availability

The code used for this work and corresponding input files are available on request to the corresponding author.

## 9. Data availability

The datasets analysed during the current study are available in the NSF NCAR Research Data Archive (RDA) repository, https://rda.ucar.edu/datasets/d628001/ for the JRA-55 data, and in the Global Land Analysis and Discovery (GLAD) laboratory at the University of Maryland repository, https://storage.googleapis.com/earthenginepartners-hansen/GFC-2023-v1.11/download.html for the Global Forest Change data.

## 10. Author contributions

All authors contributed to the study conception and design. Model conception, programmation, numerical experiments and a first version of the manuscript were performed by JC. Material preparation, data collection and data analysis were performed by MP while BM provided analytical solutions provided in the paper. CC reviewed the links between different fields: meteorology, forest and underground hydrology. All authors commented on previous versions of the manuscript, read and approved the final manuscript.

## 11. Competing Interests

The authors have no relevant financial or non-financial interests to disclose.

## 12. Acknowledgments



We thank Benjamin Belfort, Philippe Drobinsky and Florent Mouillot who provided extensive comments and suggestions on an early draft of this paper.

## 13. Funding

The authors declare that no funds, grants, or other support were received during the preparation of this manuscript.

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

responses to monsoon and precipitation. *Sci Rep* 10, 5762.