# Peer review of "More trees, more rain? The unexpected role of forest and aquifers on the global water cycle"

_EGUsphere, 2025_

## Author Comment (AC1)

Responses to HESS reviews on "More trees, more rain? The unexpected role of forest and aquifers on the global water cycle" by J. Chéry, M. Peyret, C. Champollion, and B. Mohammadi

We thank D. Sheil who provides a review of our paper, bringing comments that will be useful to improve its value. Our responses and propositions are written in red italic.

The manuscript presents a model linking forests, aquifers, and rainfall distribution. The core ideas seem new and potentially useful: presenting complex climate—vegetation feedbacks in this simple framework has possible value, especially as a conceptual tool. The positive framing of forests as sustaining inland rainfall is also constructive and welcome. However, the current paper is hard to follow (I am not sure I grasp all the details) and unsuitable for publication without revision.

**Major:**

Framing and focus – The paper claims to "test the biotic pump" but does not engage with its defining mechanism. Instead, the model explores a conceptual model of forest–aquifer–rainfall interactions. For example, the sentence "We test the biotic pump" could be replaced with "We propose a simplified model of how forests and aquifers may jointly sustain inland rainfall." *Response:*

We acknowledge that the current framing is misleading. We will revise the introduction and abstract to clarify that our model explores **forest–aquifer–rainfall interactions**, not the condensation-driven "biotic pump." We will explicitly state that our goal is to propose a **simplified conceptual framework** for understanding how forests and aquifers may jointly sustain inland rainfall, distinguishing it from previous work.

Structure, coherence and clarity – The argument is dense and difficult to follow. Assumptions, derivations, and conclusions are jumbled together. Methods and results should be separated, with explanatory text alongside equations. For instance, when presenting the conservation equations, explain each term before moving to outcomes. *Response:*

We will restructure the **Methods** and **Results** sections to improve clarity:

- *Methods:* We will separate the presentation of conservation equations and constitutive laws, explaining each term and assumption before deriving outcomes.
- Results: We will guide the reader step-by-step through the logic linking assumptions to outcomes, ensuring that statements like "forests amplify rainfall inland" are supported by clear, sequential reasoning.

Literature – The review of prior work is incomplete and risks misleading readers. For example, the manuscript cites criticisms of the biotic pump but omits published rebuttals of this criticism and various other studies that have advanced the debate. At minimum—if the focus on the biotic pump is maintained—the authors should add references to both critiques and responses, note further advances, and explain where their model and implied results sit relative to these debates. *Response:*

We agree that the literature review is incomplete. We will:

- Remove the focus on the "biotic pump" in the introduction.
- Expand the review to include **key mechanisms and models** for long-term moisture transport and moisture-rain conversion, citing both critiques and advances in the field.
- Position our model within this broader context, clarifying how it complements or diverges from existing debates.

Validation – Does the model capture real patterns and processes? We don't know. Even a simple comparison figure or discussion would help.

Response:

To address concerns about model validation, we will:

- Add a comparison figure using Australia as a case study, extending Fig. 2 to show how our model's predictions align with observed patterns (selected profiles of relative humidity and precipitations).
- Include a discussion section to interpret these comparisons, highlighting strengths and limitations of the model's predictive power.

**Specific sections:**

Introduction – Lacks a clear framing of the research gap. It blends critiques of the biotic pump with broad statements about forests and hydrology, which makes it difficult to identify a clear focus. The introduction should be restructured to identify the unresolved question and state precisely how the new model contributes.

Response:

We will restructure the introduction to:

- 1. Clearly define the **research gap**: How do forests and aquifers interact to sustain inland rainfall, and what are the limitations of current models?
- 2. Outline the **three key processes** our model addresses: moisture transport, moisture-rain conversion, and the hydrologic cycle.
- 3. End with a concise statement of our model's novelty and its potential as a conceptual tool.

Section 2 and Figure 1 – The purpose is unclear. It is not evident whether these results reproduce earlier work or simply serve as context/background. As written, the section is difficult to follow, and the connection to the rest of the paper is weak. If Figure 1 is meant to replicate established results, this should be stated explicitly and properly cited; if it introduces something new, the text should highlight what differs from prior publications and why this matters. In either case, the caption should clarify the intent. It did stimulate questions (not sure if relevant): If a comparison with past publications is intended then we need to know are the data suitable to show what is intended? How do these patterns compare if other data sources are used? What are the differences and why? *Response*:

We recognize that Section 2 and Figure 1 lack clarity and purpose. We will:

- Remove Figure 1 and its associated text to streamline the paper.
- Ensure that all remaining figures and sections are explicitly tied to the research questions outlined in the introduction.

Results – The results are presented with little explanation of the logic linking assumptions to outcomes. It is opaque where it needs to be clear. Statements such as "forests amplify rainfall inland" are asserted without the required step-by-step justification. This section should be rewritten to guide the reader through the results.

Response:

We will revise the **Results** section to:

- *Link each result* to the research questions posed in the introduction.
- Adding dedicated experiments to explore the relative influence of wind and dispersion on moisture transport, ensuring that assertions like "forests amplify rainfall inland" would require intense new developments. Interesting for a future paper.

Discussion – The discussion is underdeveloped. The authors should expand this section to situate their findings within some broader debate(s) and to clarify whether their model supports, contradicts, or simply complements the biotic pump hypothesis (a clearer goal would help too). *Response:*

We will expand the **Discussion** to:

- Situate our findings within the broader debate on forest–rainfall interactions.
- Clarify whether our model **supports**, **contradicts**, **or complements** the biotic pump hypothesis.
- Discuss the advantages and limitations of our model formulation, using the Australia case study as a point of comparison.

Conclusions – This section seems more a restatement of intent than a synthesis. Asserting that "forests sustain rainfall" seems too general (we know that already). The conclusions should restate the core contribution and new insights with clarity and precision. *Response:*

We will revise the **Conclusions** to:

- Synthesize the core contribution of our model, avoiding overly broad statements like "forests sustain rainfall."
- Highlight the **specific insights** our framework provides, such as the role of aquifers in moisture-rain conversion, and suggest directions for future research.

**Minor:**

Terminology is inconsistent; clarify early whether "biotic pump" refers to the original condensation-driven hypothesis or a broader forest-rainfall linkage.

Response:

We will restrict the use of "biotic pump" to the original condensation-driven hypothesis.

Long, complex sentences should be broken into shorter, clearer ones. *Response*:

We will do so.

Overall: contains some stimulating ideas about important topics but needs substantial work on framing, clarity, literature, and validation before it might be published. *Response*:

We will try to fulfill these recommendations in a revised version.

---

## Author Comment (AC2)

Responses to HESS reviews on "More trees, more rain? The unexpected role of forest and aquifers on the global water cycle" by J. Chéry, M. Peyret, C. Champollion, and B. Mohammadi Revewier 2 (A. Makarieva)

We thank Anastassia Makarieva (AM) who provides a careful review of our paper, bringing new elements that will be useful to improve its value. Our responses and propositions are written in red italic.

**General comments**

The authors acknowledge the complexity of plant—water interactions and aim to present a tractable model for studying the coupling between vegetation and the terrestrial water cycle. Such "mind-sized" models (see Bardi, 2013), though simplified relative to more sophisticated frameworks, can be valuable for elucidating the key processes governing the dynamics of interest—in this case, how vegetation change may influence the water cycle.

This effort aligns with the growing recognition in environmental science that, while numerical modeling has advanced to produce increasingly detailed scenarios at high spatial resolution, process-level understanding often lags behind (e.g., Byrne et al., 2024), making it difficult to judge which scenarios are realistic. Consequently, there is a need for more transparent quantitative approaches, particularly those accessible to specialists from different disciplines. In this context, the present work represents a welcome contribution to the field.

At the same time, the analysis and the model possess certain limitations that need to be addressed to ensure the paper's broader relevance and clarity.

First, for atmospheric moisture transport, the model includes—alongside horizontal advection—an additional term the authors call the "dispersion term." This term represents turbulent diffusion, which is inherently scale-dependent (a point not discussed by the authors). This formulation contrasts with previous studies of large-scale moisture transport that is dominated by advection (e.g., Trenberth et al., 2011). Meanwhile, in most numerical experiments the authors set advection to zero by assuming zero wind velocity. Such settings cannot yield meaningful insights into real-world water cycles. *Response:*

Concerning the "dispersion term", this expression is one used by scientists working on pollutants dispersion at large distances over the atmosphere (nuclear products, volcanoes products, different types of chemicals). As AM noticed, large-scale moisture transport models often neglect this term. We propose to bring some elements to explain why both advection and dispersion must be included a priori in our model. Also, we agree that zero wind velocity is not a realistic setting, despite it is useful to show the impact of dispersion alone. Therefore, we also propose to include more experiments with a non-zero wind in order to show the relative impact of both wind and dispersion on moisture and precipitation patterns.

Second, while the authors focus on precipitation and its relationship to added vegetation, this is the less controversial aspect. The more critical question, as also documented by the studies quoted by the authors, concerns how added vegetation influences atmospheric moisture transport and, consequently, steady-state runoff. A clear discussion of all terms in the water budget—and their physical and biological determinants—is required to clarify this issue, but is not provided. In addition, the results appear internally inconsistent: in steady state, over a fully forested continent, precipitation is maximized and equals evapotranspiration, yet runoff is zero despite full aquifers (e.g., Fig. 11). This would correspond, for example, to an Amazon forest without an Amazon River. *Response:*

We agree that a model displaying high precipitation and no runoff is somewhat puzzling. It needs to be said that in such a case (Fig. 11), the upper aquifer is not entirely full (50% only) because vegetation captures upper aquifer at root level. Therefore, the no-runoff state is not physically inconsistent, because runoff only occurs if the upper aquifer is full. The fact that zero runoff occurs despite high precipitation is also due to the 1D character of the model, implying that the Amazon forest would by entirely flat (slightly inclined however) with no channel allowing rivers to flow.

We propose to clarify the discussion of this subject in term of 1) water budget 2) limitation of the 1D model 3) by providing cases with runoff by including advection effect. These changes would allow the reader to connect our simplified numerical experiments with the real world (Amazon basin for example).

Finally, the terrestrial water cycle depends strongly on atmospheric dynamics, which governs the transport of moisture from the ocean onto land. Consequently, the influence of vegetation on the water cycle depends on how vegetation affects atmospheric circulation. In the present model, however, the two parameters representing atmospheric dynamics—turbulent diffusivity ( $D_h$ ) and wind speed—are specified independently of vegetation and all other parameters. By design, such a model cannot capture potential couplings between vegetation, atmospheric water vapour, and circulation dynamics. These interactions, beyond the biotic pump mechanism discussed by the authors, have been explored in other studies that are not referenced in the paper.

We think that the influence of vegetation on the water cycle depends on how vegetation affects atmospheric circulation, but also on the repartition of the water budget by evapotranspiration and aquifer pumping (which is a more classical view).

We therefore propose to present this issue on the introduction, briefly classifying the different kinds of approaches, in order to define the position of our model with respect to existing models/mechanisms.

Overall, I would recommend revising the manuscript to (i) explicitly discuss the limitations of their model and its implications for real water cycles, (ii) clarify or resolve the identified inconsistencies in their results, or, as a more profound restructuring, (iii) consider ways to incorporate possible couplings between vegetation and atmospheric dynamics.

**Response:**

Response:

We propose to work of the points (i) and (ii) in the paper core (introduction-methods-results) and to discuss (iii) in a discussion/perspective section. Indeed, the implementation or other coupling processes (ie, modeling biotic pump mechanism) would require an updated numerical formulation.

**Specific comments**

**• Equation (1)**

This equation is central to the model, as it describes atmospheric moisture transport. The authors treat the atmosphere as a single one-dimensional slab. In addition to the advection of moisture (i.e., transport by wind), the equation includes what the authors call a dispersion term, proportional to the second derivative of vertically integrated moisture content with respect to horizontal distance. This term represents turbulent diffusion, characterized by turbulent diffusivity  $D_h$ .

To justify this formulation, the authors refer to "a similar formulation" by Rasmussen (1968). However, Rasmussen's equation for vertically integrated moisture content does not include a diffusion term. Nor is such a term present in more recent and widely used formulations of the atmospheric moisture budget, for example those of Trenberth et al. (2011, their Eq. 1), Dominguez et al. (2020, their Eq. 3), or Luo et al. (2022, their Eq. 1).

**Response:**

Indeed, the Rasmussen approach was not the right citation. Thanks to a paper provided by the reviewer (Dominguez et al. 2020), we found papers (Schaeffli et al. 2012, eq. 2 p1865; Savenije 1995 eq.12 p65) that specifically include the diffuse term, but explaining that it should be neglected for the purpose of moisture transport.

We propose to include these references, and explain why the hypothesis of neglecting dispersion term may be not correct for some settings. This is a point that clearly needs attention as this may influence the moisture spreading at large scale, as it has been demonstrated for chemicals transport and spreading in the atmosphere (eg. Pisso et al. 2006).

The references cited by the authors on the propagation of passive tracers—where turbulent diffusion can indeed be significant—do not justify including a diffusive term in the large-scale moisture budget. *Response:*

We noticed that many studies treating of various chemicals products (gaseous or small sized particles) incorporate the diffusive term. We don't see a priori why this term would be important for chemical transport modeling and be negligible for moisture transport (that is a kind of chemical product among others), as the spatial and temporal scales are similar. We propose to discuss this point in the revised version, as it appears to be a novel approach for the purpose of moisture spreading in the atmosphere.

Consider a case in which the annual-mean wind over land is zero, but its direction reverses seasonally. A nonzero annual-mean moisture transport can still arise from seasonal winds (e.g., monsoons) through the covariance of wind and humidity anomalies (see, e.g., Eq. 4 of Luo et al. 2022). However, this transport by large-scale variability cannot be parameterized as local turbulent diffusion in the authors' formulation. It remains dominated by advective moisture transport, which can proceed against local moisture gradients. For example, in hurricanes, moisture is advected toward the eyewall, where the vertically integrated moisture content reaches its maximum.

I therefore recommend revising the formulation of atmospheric moisture transport and modifying the numerical experiments accordingly, as a substantial portion of the current calculations is performed under conditions of zero advection, with transport represented solely by diffusion.

**Response:**

We recall here that a model is simplified, with one layer representing the whole atmosphere, with no horizontal wind divergence. This implies that the wind velocity is spatially constant (but may temporally vary) in order to satisfy mass conservation. Therefore, in this 1D model, wind only translates the moisture without altering its distribution. We think that our moisture conservation equation is mathematically correct (equivalent to eq. 2 of Schaeffli et al., 2012). However, we also think that neglecting the dispersion term could be incorrect in some cases. Indeed, if the average wind is small, the dispersion term becomes important for moisture transport.

We therefore propose to present our model in reference to Schaeffli et al., 2012, in order to point out the similarity and the differences between the two formulations.

**• Equation (12)**

This issue is directly related to the above. Equation (12) is used to derive a steady-state distribution of water-vapor concentration versus distance from the ocean, and the manuscript states that Eq. (12) follows from Eq. (1). However, a key term appears to have been omitted—namely, the horizontal derivative of the horizontal wind that is present in Eq. (1). This term is crucial for atmospheric moisture transport: horizontal wind speed typically declines where air ascends and hence generates precipitation. For example, a conspicuous decline in horizontal wind is observed across the Amazon—Atlantic coastal zone near the surface, with the opposite pattern at 500 hPa.

Given this critical omission, results derived from Eq. (12) and presented in the discussion, should be re-evaluated.

**Response:**

Because we consider a 1D model, the wind is constant and its spatial derivative is zero. A similar formulation is made by Schaeffli et al. 2012, and this corresponds to the mass conservation assumption. The case proposed by the reviewer would require a formulation with 2 or more superposed horizontal layers with an exchange coefficient controlling the vertical moisture motion. We propose to present more clearly our model hypotheses and discuss the intrinsic limitations of our formulation.

**• Differential impact of vegetation on different aspects of the water cycle**

The paper's title refers to the "unexpected role of forests" in the "global water cycle," yet the discussion does not address a critical point: vegetation can influence different components of the moisture budget in distinct ways, notably precipitation and runoff.

The authors state that the effect of large forests on "regional pluviometry" remains obscure (line 99). However, the vast majority of global climate models agree that deforestation reduces precipitation, both regionally (over deforested areas) and globally (e.g., Table 3 in Luo et al., 2022). This robust result also follows from first principles: globally, evapotranspiration balances precipitation, so an increase in evapotranspiration over large areas must lead to a corresponding increase in global precipitation.

What remains much less clear is the impact of vegetation on atmospheric moisture transport. Here, model responses vary greatly in both sign and magnitude, with no robust consensus (see the same table in Luo et al., 2022).

In this context, based on their formulation involving turbulent diffusion, the authors obtain a solution in which a fully forested continent has zero runoff (Fig. 11). This outcome arises because, in the model, the only effect of vegetation is to increase evapotranspiration, which raises water vapor content over land and thereby reduces the land—ocean moisture gradient—and hence the diffusional transport that the authors (incorrectly) assume to be a possible dominant mechanism of atmospheric moisture transfer. It can be shown that runoff reduction will invariably accompany increased evapotranspiration unless atmospheric circulation changes (e.g., Makarieva et al. 2025). Because the circulation (represented through the turbulent diffusion coefficient) remains fixed in the authors' model, the result is zero runoff when evapotranspiration is maximized over a fully forested continent.

Given that this outcome does not align well with observations—as major forests are typically associated with major river systems—such a result warrants a substantially deeper discussion to clarify its physical meaning and implications.

**Response:**

We agree with the considerations above, and we propose to include some numerical experiments showing the relation between moisture advection, dispersion, evapotranspiration and the onset of river systems, and also a related discussion.

**• Zero runoff at full aquifers**

While runoff can theoretically vanish when precipitation (P) equals evapotranspiration (ET)—that is, when all precipitating water re-evaporates—it cannot be zero if the aquifers are full. Because land lies above sea level, gravitational flow necessarily drives groundwater discharge toward lower elevations and ultimately to the ocean.

Figure 11 shows ET=P, zero runoff, and full aquifers over a forested continent. Unless this outcome results from a misunderstanding (which should be clarified), it points to a physically inconsistent behavior of the model, most likely originating from its treatment of groundwater storage and discharge. This inconsistency should be explicitly addressed.

\*Response:

As it appears on Fig. 11d, the upper aquifer is only 50% full due to water removal by forest. Therefore, overflow does not take place and no runoff occurs. By contrast, lower aquifer overflows to the upper aquifer, that is physically consistent from a hydrologic viewpoint. Also, we are numerically checking that the budget of water mass transport is closed at each time step, proving that our model is physically consistent.

We propose to better state these points in the numerical experiment section.

**• Vegetation impact on atmospheric dynamics**

Since land inevitably loses water to the ocean, rainfall over land cannot persist without an atmospheric moisture transport from the ocean. Without such transport, precipitation could arise only from local evaporation. However, sustaining local evaporation requires a terrestrial water store, which will gradually drain to the ocean under gravity. Once that store is depleted, both evaporation and precipitation will cease. Thus, continuous rainfall over land fundamentally depends on a steady advective inflow of atmospheric moisture from the ocean.

Because this inflow is governed by advection (i.e., winds), a central question regarding the influence of vegetation on the water cycle is how vegetation may affect atmospheric circulation itself. In the authors' model, however, this influence cannot be examined in principle, since the only parameter representing air circulation—wind speed—is prescribed independently of any vegetation characteristics.

For this reason, while the authors refer to the biotic pump concept on several occasions, their model, by design, cannot explore it: the biotic pump mechanism is based on a positive feedback between vegetation and atmospheric moisture transport. In addition to the biotic pump mechanism, other researchers have proposed alternative pathways linking enhanced condensation of water vapor to increased atmospheric moisture transport. (For a recent overview of the historical development of these ideas, see Makarieva et al., 2025.)

I suggest that the authors explicitly address this issue and decide whether they wish to extend their model to incorporate such coupling. If atmospheric circulation remains a prescribed (free) parameter, this limitation should be clearly acknowledged and discussed.

**Response:**

The reviewer is entirely right, and we will address this issue by exposing in the introduction the different ways by which water vapor can be horizontally transported: pure advection (meteorology), forest induced advection (biotic pump mechanism), horizontal dispersion due to self-mixing into the atmosphere. References provided by AM will be useful to do so.

**• Miscellaneous**

The authors begin their paper by analyzing precipitation distribution along transects originally studied by Makarieva and Gorshkov (2007) introducing the biotic pump. There were several other studies exploring similar dependencies, in particular, Poveda et al. (2014) and Molina et al. (2019). Notably, when the spatial distribution is exponential (i.e., rainfall exponentially declines with distance from the ocean), taking its average over a large distance is not informative, as this hides the fact that due to the rapid decline precipitation can be extremely low over most part of the continent. *Response:*

We will consider the other studies mentioned by the reviewer, and we will present a better summary of the spatial distribution of precipitations.

Role of vegetation decline for coastal precipitation decrease in Australia (relevant to the authors' Fig. 2) was explored by Andrich and Imberger (2012).

Response:

We thank the reviewer for this relevant publication that we will use for our introduction and discussion

In their Eq. (4), the authors provide an expression for local precipitation that depends on the n-th power of relative humidity. On line 285, "(ref?)" presumably refers to a source for this formulation. Please provide the missing reference and/or discuss how n in Eq. (4) is justified. *Response:*

This is indeed an omission and we will provide adequate references and explanations for setting eq. 4 and n value.

It would be helpful to provide a summary table listing all numerical experiments along with their specific parameter values and settings.

Response:

We will provide a summary table in a revised version.

The model code should be uploaded to a publicly accessible repository (e.g., Zenodo or GitHub), rather than requiring interested readers to request it from the corresponding author. *Response:*

That is indeed a good suggestion, we will proceed this way, adding also a short user's guide.

Please define "cumulated" – what is cumulated precipitation and runoff? *Response*:

We will define this term (corresponding to the time integration of precipitation and runoff rate over one year).

**Conclusions**

From my perspective, tractable, "mind-sized" models such as the one presented by the authors are exceptionally valuable, as they enable meaningful discussion of the fundamental processes underlying the studied phenomena. Indeed, the present critical comments were possible precisely because the model is transparent and accessible. Besides, the authors offer an interesting discussion of multiple interconnected issues spanning diverse disciplines, including aspects of past climate change.

I therefore encourage the authors to revise their formulation, re-evaluating the role of moisture advection, resolving the runoff inconsistency, and explicitly discussing atmospheric moisture transport. In a revised form, this model could make a substantial contribution to the broader community of scientists investigating the terrestrial water cycle.

We will address the concerns raised above in the revised version.

**References**

Andrich, M. A., & Imberger, J. (2013). The effect of land clearing on rainfall and fresh water resources in Western Australia: a multi-functional sustainability analysis. International Journal of Sustainable Development & World Ecology, 20(6), 549-563. https://doi.org/10.1080/13504509.2013.850752

Bardi, U. (2013). Mind sized world models. Sustainability, 5(3), 8 https://doi.org/10.3390/su503089696-911

Byrne, M.P., Hegerl, G.C., Scheff, J. et al. Theory and the future of land-climate science. Nat. Geosci. 17, 1079–1086 (2024). https://doi.org/10.1038/s41561-024-01553-8

Dominguez, F., Hu, H., & Martinez, J. A. (2020). Two-layer dynamic recycling model (2L-DRM): Learning from moisture tracking models of different complexity. Journal of Hydrometeorology, 21(1), 3-16. https://doi.org/10.1175/JHM-D-19-0101.1

Luo, X., Ge, J., Guo, W., Fan, L., Chen, C., Liu, Y., & Yang, L. (2022). The biophysical impacts of deforestation on precipitation: results from the CMIP6 model intercomparison. Journal of Climate, 35(11), 3293-3311. https://doi.org/10.1175/JCLI-D-21-0689.1

Makarieva, A. M., Nefiodov, A. V., Cuartas, L. A., Nobre, A. D., Andrade, D., Pasini, F., & Nobre, P. (2025). Assessing changes in atmospheric circulation due to ecohydrological restoration: how can global climate models help?. Frontiers in Environmental Science, 13, 1516747. https://doi.org/10.3389/fenvs.2025.1516747

Molina, R. D., Salazar, J. F., Martínez, J. A., Villegas, J. C., & Arias, P. A. (2019). Forest-induced exponential growth of precipitation along climatological wind streamlines over the Amazon. Journal of Geophysical Research: Atmospheres, 124(5), 2589-2599. https://doi.org/10.1029/2018JD029534 Poveda, G., Jaramillo, L., & Vallejo, L. F. (2014). Seasonal precipitation patterns along pathways of South American low-level jets and aerial rivers. Water Resources Research, 50(1), 98-118. https://doi.org/10.1002/2013WR014087

Trenberth, K. E., Fasullo, J. T., & Mackaro, J. (2011). Atmospheric moisture transpor636ts from ocean to land and global energy flows in reanalyses. Journal of climate, 24(18), 4907-4924. https://doi.org/10.1175/2011JCLI4171.1